# Long-term relative decline in evapotranspiration with increasing runoff on fractional land surfaces

**Ren Wang[1,2,3], Pierre Gentine[4,5], Jiabo Yin[6], Lijuan Chen[1,2,3], Jianyao Chen[7,8], and Longhui Li[1,2,3]**

[1]Key Laboratory of Virtual Geographical Environment (Nanjing Normal University), Ministry of Education, Nanjing, 210023, China

[2]School of Geographical Sciences, Nanjing Normal University, Nanjing, 210023, China

[3]Jiangsu Center for Collaborative Innovation in Geographical Information Resource Development and Application, Nanjing, 210023, China

[4]Earth and Environmental Engineering Department, Columbia University, New York, NY 10027, USA

[5]Earth Institute, Columbia University, New York, NY 10025, USA

[6]State Key Laboratory of Water Resources and Hydropower Engineering Science, Wuhan University, Wuhan, 430072, China

[7]School of Geography and Planning, Sun Yat-sen University, Guangzhou, 510275, China

[8]Guandong Key Laboratory for Urbanization and Geo-simulation, Sun Yat-sen University, Guangzhou, 510275, China

*Correspondence to*: Ren Wang (wangr67@mail2.sysu.edu.cn) or Pierre Gentine (pg2328@columbia.edu)

**Abstract.** Evapotranspiration (ET) accompanied by water and heat transport in the hydrological cycle is a key component in regulating surface aridity. Existing studies documenting changes in surface aridity have typically estimated ET using semi-empirical equations or parameterizations of land surface processes, which are based on the assumption that the parameters in the equation are stationary. However, plant physiological effects and its responses to a changing environment are dynamically modifying ET, thereby challenging this assumption and limiting the estimation of long-term ET. In this study, the latent heat flux (ET in energy units) and sensible heat flux were retrieved for recent decades on a global scale using machine learning approach and driven by ground observations from flux towers and weather stations. This study resulted in several findings, that is, the evaporative fraction (EF)—the ratio of latent heat flux to available surface energy—exhibited a relatively decreasing trend on fractional land surfaces; in particular, the decrease in EF was accompanied by an increase in long-term runoff as assessed by precipitation (P) minus ET, accounting for 27.06% of the global land areas. The signs are indicative of reduced surface conductance, which further emphasizes that surface vegetation has major impacts in regulating water and energy cycles, as well as aridity variability.

# 1 Introduction

Evapotranspiration (ET) mainly includes two processes: (1) evaporation from soil and plant surfaces and (2) transpiration from plants to the atmosphere (Miralles et al., 2020). These processes connect the transfer of moisture and energy in soil, vegetation, and atmospheric systems (Salvucci et al., 2013; Yang et al., 2020). Quantifying changes in the exchange of moisture and heat between the land and atmosphere is very important for understanding and characterizing water and energy cycles, which has implications in various fields such as hydrology, climatology, and agronomy (Hoek van Dijke et al., 2020; Gentine et al., 2016; Komatsu and Kume, 2020).

ET is expected to intensify with the warming climate, thereby contributing to the increase in surface aridity stress (Baruga et al., 2020; Berg et al., 2016; Fu et al., 2014; Trenberth et al., 2014). However, quantification of changes in aridity/wetness is usually derived from traditional drought indices such as the Standardized Precipitation Evapotranspiration Index (Vicente-Serrano et al., 2015), which is embedded with a semi-empirical equation, such as the Thornthwaite equation or Penman–Monteith equation, for ET estimation (Dai et al., 2013; Sheffield et al., 2012). Using potential evaporation rather than actual ET or calculating offline ET using meteorological variables from climate model outputs in traditional drought indices, the calculation implicitly assumes that soil can always supply moisture to meet the atmospheric evaporation demand, which is an incorrect assumption for most land surfaces (Greve et al., 2014; Milly and Dunn, 2016; Yang et al., 2020). Moreover, when using a semi-empirical equation for ET estimation, some parameters such as soil surface resistance and stomatal resistance, are assumed to be stationary over time; however, we know that these parameters are dynamically changing with environmental conditions (Miralles et al., 2011; Yang et al., 2019; Zhou et al., 2016).

Why are the soil surface resistance and stomatal resistance not stationary? Changes in plant stomata and leaf area, with increasing $CO_2$ concentrations in particular, reshape the allocation of surface energy and affect plant transpiration (Forzieri et al., 2020; Sorokin et al., 2017; Mallick et al., 2016; Williams and Torn, 2015). With increasing $CO_2$ concentrations, the density and opening degree of leaf stomata decrease, while the water-use efficiency and biomass production of plant increase, which can modify vegetation transpiration and even affect soil moisture or surface runoff (Keenan et al., 2013; Massmann et al., 2019; Orth and Destouni, 2018; Rigden et al., 2016; van Der Sleen et al., 2015; Wagle et al., 2015). Vegetation transpiration occupies most of ET amount, so vegetation control effects can greatly alter the variability of land surface ET (Costa et al., 2010; Jaramillo et al., 2018; Wei et al., 2017; Williams et al., 2012). Moreover, human activities including agricultural irrigation and land use management, are constantly altering the exchange of water and heat between terrestrial ecosystems and the atmosphere (Padrón et al., 2020; Teuling et al., 2019). When these effects are taken into account, the semi-empirical equations for estimating ET and traditional drought indices also face challenges (Yang et al., 2020). Existing studies with respect to global surface fluxes inferred from flux tower observations, remote sensing products, and reanalysis data, e.g., the surface fluxes driven by model tree ensemble (Fluxnet-MTE), rely on the satellite era and instantaneous meteorological observations (Jung et al., 2010; Jung et al., 2011; Miralles et al., 2013). Thus, the existing products cannot be

used for long-term trends as they cannot represent the long-term effects of confounders such as $CO_2$ or species composition changes. This is why we use an opposite view – we use in essence a boundary layer energy budget (Salvucci and Gentine, 2013; Gentine et al., 2016) except that we lump non-linear effects of changing environment factors on surface energy fluxes in a neural network. Indeed, the diurnal cycle of temperature is directly related to sensible heat flux, and the course of specific humidity related to the rate of latent heat flux variation (Gentine et al., 2011). If there are changes in latent heat flux due to vegetation in response to higher $CO_2$, this is still captured by the change in the specific humidity.

In this study, we propose a new strategy for estimating latent heat flux ($\lambda E$) (ET in energy units) and sensible heat flux (H) using machine learning approach and ground observations from flux towers and weather stations. This strategy utilizes daily observations of meteorological variables such as temperatures, humidity, and solar radiation. A major advantage of such retrieval is that it does not rely on any assumption on a $CO_2$ effect on the link between environmental variables and fluxes. Indeed, we flipped the strategy around its head by diagnosing the diurnal changes in temperature and humidity in the boundary layer. As such this diurnal cycle reflects naturally any change in $CO_2$. For instance, if stomata were to substantially close they would increase H and reduce $\lambda E$. This would in turn lead to increased temperature diurnal range and reduced air humidity in the boundary layer (Salvucci and Gentine 2013, Gentine et al. 2016). Therefore, this $CO_2$ effect is completely detectable. This is a major advantage of our method based on a boundary layer energy budget, as the physics of the boundary layer does not change (fluid dynamics). Moreover, the observational record of the weather station network is not only longer, but also extends to more remote places, such as the tropics. This study also employed the evaporative fraction (EF), i.e., the ratio of $\lambda E$ to the sum of $\lambda E$ and H, and a proxy for long-term runoff, i.e., the difference of precipitation (P) and ET (P−ET), to quantify the change in aridity/wetness.

## 2 Observational data and methodology

### 2.1 Flux tower observational data

We collected the half-hourly/hourly observational data and the integrated daily product from the FLUXNET2015 FULLSET dataset (Pastorello et al., 2020). To control the quality of the observational dataset, this study only used measurements and good-quality gap-filled data from 212 globally distributed flux towers (Supplementary Fig. S1a). The flux towers used in this study across various climate regions and land cover types (Fig. 1). The longest period of data availability is 22 years. This study intended to build a machine learning model for retrieving latent heat and sensible heat fluxes on a daily scale. Therefore, daily-scale data of top-of-atmosphere shortwave radiation, vapor pressure deficit (VPD), mean temperature, and surface wind speed were collected from the integrated daily product. VPD was used to calculate relative humidity. Daily maximum and minimum temperatures were obtained from the half-hourly/hourly flux tower measurement data. Moreover, daily-scale $\lambda E$ and H were also collected from the integrated daily product. The underlying surfaces of the flux towers covered different plant function types (PFTs). According to the classification scheme of International Geosphere-Biosphere

Programme, the PFTs include Croplands (CRO), Deciduous Needleleaf Forests (DNF), Evergreen Needleleaf Forest (ENF), Evergreen Broadleaf Forest (EBF), Deciduous Broadleaf Forest (DBF), Mixed Forest (MF), Grasslands (GRA), Savannas (SAV), Woody Savannas (WSA), Closed Shrublands (CSH), Open Shrublands (OSH), Wetlands (WET), and Snow and Ice (SNO). These flux tower observation data across different ecosystems are used to train machine learning model for predicting latent heat and sensible heat fluxes. The global $CO_2$ fertilization effects have changed over recent decades (Wang et al., 2020), and thus the observation period of the Fluxnet data are long enough to capture $CO_2$ effects on vegetation.

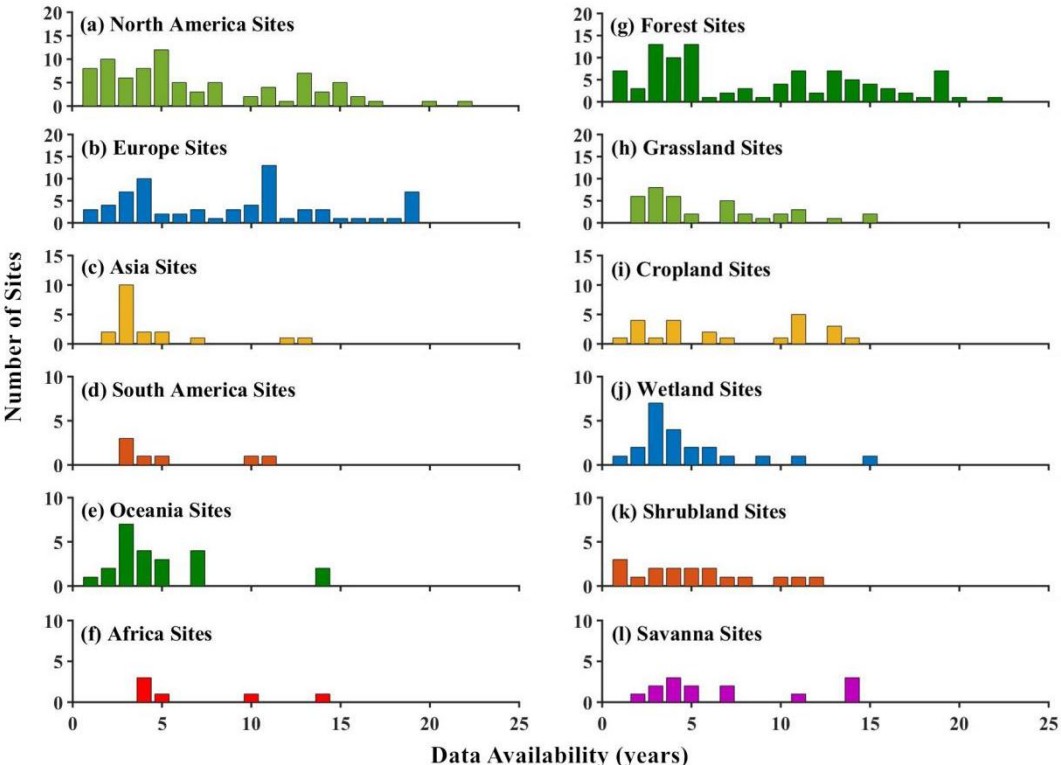

**Figure 1.** Data summary of the flux towers used in this study.

## 2.2 Weather station observation data

Daily observational records of precipitation (P), temperature (mean, maximum, and minimum temperatures), dew point temperature, and wind speed at weather stations were collected from the Global Summary of the Day (GSOD) during the 1950–2017 period. Dew point temperature data were used to calculate the relative humidity, and the daily weather station data on the global land were used to drive a well-trained machine learning model to retrieve surface fluxes. The quality of the data was controlled through several procedures (Durre et al., 2010; Matsuura et al., 2009; Yin et al., 2018). First, we divided the weather stations into two groups: the original stations and the target stations. We used 20 048 sites in total as the original station group (Supplementary Fig. S1b). The target station group was obtained according to the following steps: (1)

The stations with a time series spanning less than ten years in length were excluded; (2) if the stations had the same geographic coordinates, we used the stations with a long observation record to replace the stations with a short observation record; (3) if there were multiple stations having different coordinates in a 0.1-degree grid, we removed the stations with a short observation record. After filtering, the target station group which were determined to be used to estimate long-term trends were obtained.

Other procedures for controlling data quality were also implemented. Any implausible values, such as negative precipitation or maximum temperature lower than the minimum temperature on that day, were excluded. Monthly mean, maximum, and minimum temperatures, as well as monthly precipitation were derived from daily observational data at the original stations. Considering the large uncertainty in the observational data of precipitation, we also compiled the daily precipitation records with precipitation records in another archives, i.e., the Global Historical Climatology Network (GHCN-Daily). The daily records of weather stations in the GSOD that had the same coordinates as the GHCN-Daily were compared, and the missing daily records were supplemented using the GHCN-Daily archives. Monthly precipitation, temperatures (mean, maximum, and minimum temperatures), relative humidity, and surface wind speed were calculated when the number of missing days within a month was no more than seven days. Additionally, missing monthly data from the target stations were spatially interpolated from the original weather stations using the Kriging method.

## 2.3 Top-of-atmosphere shortwave radiation model

Solar shortwave radiation is a key factor affecting surface energy and water cycles. Since there is no reliable long-term observational solar radiation data, this study uses shortwave radiation at the top of the atmosphere (top-of-atmosphere shortwave radiation) as a replacement. Cloud effects are inherently captured by the diurnal cycle of temperature and humidity (Gentine et al., 2013a,b). Daily top-of-atmosphere shortwave radiation converted from the hourly top-of-atmosphere shortwave radiation was forced to drive the model for predicting the daily $\lambda E$ (H) at the target weather stations. The amount of incoming shortwave radiation at any location/time at the top of atmosphere is a function of Earth–Sun geometry, which is defined as: i) latitude (i.e., location); ii) hour of day (due to the rotation of the earth); and iii) day of year (due to the tilted axis of the earth and its elliptical orbit around the sun). Several models for the top-of-atmosphere fluxes based on these inputs are available at varying levels of precision. The time-location model (Margulis, 2017) used in this study is shown as follows.

$$R_{s0} = \begin{cases} I_0 \dfrac{\cos \theta_0}{d^2}, & daytime : |\theta_0| \le 90^\circ \\ 0 \;\;, & nighttime \end{cases} \tag{1}$$

where the cosine of the solar zenith angle is as follows:

$$\cos \theta_0 = \sin \delta \sin \lambda + \cos \delta \cos \lambda \cos \tau \tag{2}$$

$$\delta = \frac{23.45\pi}{180} cos\left[\frac{2\pi}{365}(172-DOY)\right] \tag{3}$$

$$\tau = 2\pi \frac{T_h-12}{24} \tag{4}$$

$$d = 1+0.017cos\left[\frac{2\pi}{365}(186-DOY)\right] \tag{5}$$

Here, $\theta_0$ is the solar zenith angle, $\delta$ is the declination angle, $\lambda$ is latitude, $\tau$ is the hour angle, $DOY$ represents the day of year, $d$ represents the distance between the sun and Earth normalized by the mean distance, and $T_h$ represents solar hour.

## 2.4 Artificial neural network model training

The artificial neural networks (ANN) have been shown to be powerful non-linear regression algorithm, and unlike other machine learning algorithms, ANN can build multi-layer and multi-node network models to achieve deep learning of a
complex simulation. Pure ANN model has been proven to show good performance in retrieving surface fluxes (Chen et al., 2020; Haughton et al., 2018; Zhao et al., 2019). In this study, we trained a multi-layer feedforward neural network model that consisted of an input layer, hidden layers, and an output layer to predict daily λE and H at the globally distributed weather stations. To identify the sensitivities of latent heat and sensible heat fluxes to different variables in the retrieval, we used different variable combinations to train ANN model and tested the changes in the model performance (Supplementary
Table S1). Top-of-atmosphere shortwave radiation, relative humidity, wind speed, and the mean, maximum, and minimum temperatures were determined to be the inputs of the neural network (Supplementary Table S2).

In the process of training ANN model, input data were randomly divided into three subsets using the percentages of 80%, 10%, and 10% for training, validation, and testing, respectively. Mean squared error (MSE) was used to evaluate the
performance of the neural network in the training process of adjusting weight. Root mean squared error (RMSE) and Pearson correlation coefficient (R) between the ANN predicted λE (H) and the observed λE (H) in the validation set were used to evaluate the retrieval performance of the well-trained ANN model. A neural network with 2 hidden layers can achieve the same performance as with a large number of hidden layers, so we used the lowest complexity model and enhanced its nonlinear ability by adding neurons. As for the optimal number of neurons, we initially tested it according to an empirical
formula, i.e., $h = \sqrt{(n+m)} + a$ ($n$ is the number of input neurons, $m$ is the number of output neurons, and $a$ is a constant ranging from 0 to 10). This empirical formula can provide a reference for us to choose the number of neurons when training neural network, and it can reduce the possibility of overfitting. The neural network was determined to have two hidden layers and 15 neurons per hidden layer, and the ANN model showed good performance and appropriate training time (Supplementary Fig. S2). A tangent sigmoid transfer function was used in the hidden layers, and a linear transfer function
was used in the output layer. To avoid over-fitting, the early stopping method was used, that is, we recorded the best

validation accuracy during the training process, and the training was stopped when the MSE was no longer reduced after going through additional epochs. The maximum number of training epochs and training accuracy goal were set to 500 epochs and 0.0001, respectively. Once one of the parameters exceeded the threshold, the training was stopped.

## 2.5 EF linked to surface resistance ($r_s$) and aerodynamic resistance ($r_a$)

Here, we show that a long-term decline in EF can be strongly impacted by an increase in surface resistance ($r_s$). The latent heat flux ($L_v E$) is expressed by the formula:

$$L_v E = L_v \rho \frac{e_{sat}(T_s) - e_a}{r_a + r_s} \tag{6}$$

where $L_v$ is the latent heat of vaporization, $E$ is evaporation flux, $\rho$ is air density, $T_s$ is near-surface air temperature, $e_{sat}(T_s)$ is saturated vapor pressure at the surface, $e_a$ is actual vapor pressure, $r_a$ is aerodynamic resistance, and $r_s$ is

surface resistance. EF can be expressed as follows.

$$EF = \frac{L_v E}{L_v E + H} = \frac{L_v \rho \dfrac{e_{sat}(T_s) - e_a}{r_a + r_s}}{L_v \rho \dfrac{e_{sat}(T_s) - e_a}{r_a + r_s} + H} \tag{7}$$

We used the linearized Clausius–Clapeyron relation (Eq. (8) and Eq. (9)) to simplify Eq. (7).

$$e_{sat}(T_s) = e_{sat}(T_a) + \Delta(T_s - T_a) \tag{8}$$

$$T_s - T_a = \frac{H}{\rho c_p} \tag{9}$$

Where $T_a$ is the air temperature, $e_{sat}(T_a)$ is saturated vapor pressure of the air, and $\Delta = \dfrac{L_v}{R_v} \dfrac{e_s}{T^2}$, $R_v$ is the gas constant for water vapor. Furthermore, $c_p$ is the specific heat capacity, which is 4216 J kg$^{-1}$ K$^{-1}$ when the temperature is 0 °C.

$$EF = \frac{\dfrac{L_v \rho}{r_a + r_s}((e_{sat}(T_a) - e_a)) + \dfrac{\Delta H}{\rho c_p}}{\dfrac{L_v \rho}{r_a + r_s}((e_{sat}(T_a) - e_a)) + H} \tag{10}$$

$$= \frac{\dfrac{L_v \rho}{r_a + r_s} \left\{ VPD + \dfrac{\Delta H}{\rho c_p} \right\}}{\dfrac{L_v \rho}{r_a + r_s} \left\{ VPD + \dfrac{\Delta H}{\rho c_p} \right\} + H} \tag{11}$$

$$= \cfrac{1}{1+\cfrac{r_a+r_s}{L_v\rho(\cfrac{\Delta}{\rho c_p}+\cfrac{VPD}{H})}} \qquad (12)$$

The incremental variation of $\frac{VPD}{H}$ is small because both variations of VPD and H are proportional to the temperature variation. EF can be expressed as follows:

$$\frac{1}{EF}=1+\frac{r_a+r_s}{\cfrac{L_v}{c_p}\Delta} \qquad (13)$$

Hence, $r_s$ is a function of EF.

$$r_s=\frac{L_v}{c_p}\Delta(\frac{1}{EF}-1)-r_a \qquad (14)$$

Annual EF ranges from 0 to 1, and EF is closely connected with surface resistance and aerodynamic resistance. A decline in EF can be induced by an increase in surface resistance. $r_a$ is a function of wind speed, and the variation in $r_a$ is relatively small while the variations in $r_s$ can be strong.

## 3 Results and discussion

### 3.1 ANN model retrievals

Cross-validations of the ANN model were performed in terms of values and trends. We randomized samples of 10 randomly chosen flux towers from different PFTs as the validation set, and then used the remaining samples to train the ANN model. The predicted daily λE (H) values of the validation set were compared with their observed values (Fig. 2a). The R between predicted daily λE and observed daily λE is 0.849 and the R between predicted daily H and observed daily H is 0.743, and both correlations are significant at the p<0.001 level. Moreover, we trained a random forest (RF) model for predicting daily λE and H based on the same Fluxnet2015 dataset as the ANN model. The RF model shows very similar performance to the ANN model. The correlation coefficients of RF model in predicting daily λE and daily H are 0.777 (p<0.001) and 0.756 (p<0.001), respectively (Supplementary Fig. S3). Therefore, it is feasible to use the neural network algorithm to retrieve surface fluxes. Cross-validations were also performed in different land covers (Supplementary Fig. S4). The abilities of the trained ANN model for predicting latent heat and sensible heat fluxes were different for various PFTs. With the exception of OSH (R=0.680, p<0.05), the R of daily λE of DBF, MF, SAV, GRA, CRO, and WET were all greater than 0.80, and all correlations were significant at the p<0.001 level. A common feature of these PFTs is that they belong to the ecosystems with relatively open water bodies or high vegetation coverage, while the OSH is mixed with vegetation and bare soil and

thus the vegetation coverage is highly heterogeneous. Therefore, the R at OSH was relatively low (R=0.680), but the correlation was significant at the p<0.05 level. With respect to daily H, the correlation coefficients for all PFTs were greater than 0.716 with the exception of R for CRO (R=0.656, p<0.05), and all were statistically significant at the p<0.001 level. In addition, the trained ANN model also shows good simulation ability under some other ecosystems with relatively sparse vegetation cover such as savannas (SAV), grasslands (GRA), croplands (CRO), and wetlands (WET) (Supplementary Fig. S5). In summary, in addition to OSH, the accuracy of retrieving λE is relatively high in GRA, CRO, WET, and various forest ecosystems, and these ecosystems were characterized by sufficient water supply or dense vegetation. For the estimation of H, except for the estimation of H in GRA, the correlations of predicted and observed H at all ecosystems are correlated at the p<0.001 level, especially in forest. It needs to be emphasized that the magnitude of R could be affected by the number of samples, and the sample number in those cross-validations are large. As for the prediction of trends in λE and H, the ANN model also shows good performance (Fig. 2b). All correlation coefficients between the estimated λE (H) trends and the observed λE (H) trends exceeded 0.90 (p<0.001) over ENF, DBF, GRA, and WET, and the correlations over MF, OSH, and CRO exceeded 0.80 (p<0.001), and the correlations are greater than 0.70 (p<0.001) in EBF and SAV (Supplementary Fig. S6 and Fig. S7). In most cases, the estimations of λE (H) trends are more reliable than the retrieved λE (H) values.

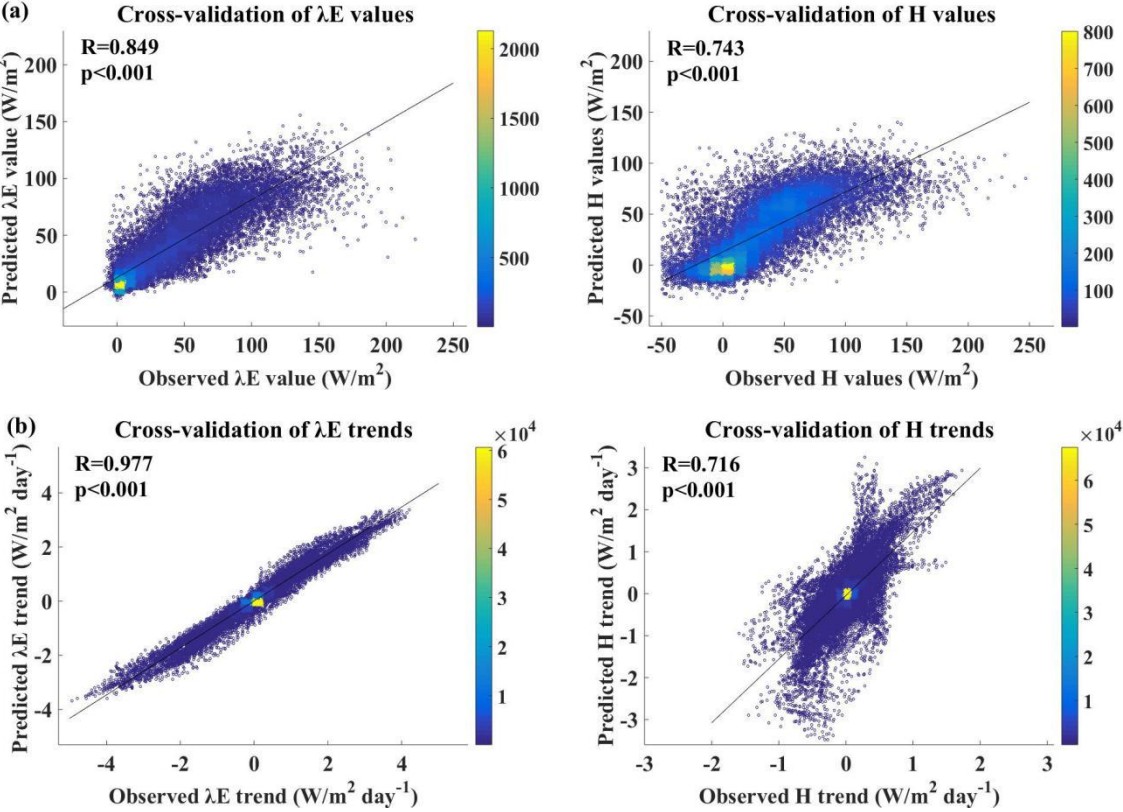

**Figure 2.** Density scatter plot for (a) the cross-validation in terms of values and (b) the cross-validation in terms of trends. The validation set of values cross-validation is composed of 10 flux towers randomly selected from different plant function

types, and the validation set of trends cross-validation is composed of the trends calculated from all time periods of the availability of the flux tower observations. The trends are calculated using linear trend estimation.

The uncertainty and bias characteristics of the ANN model retrievals were further analyzed on both daily and monthly scales. At the daily scale, the RMSE of λE (H) ranged from 26.05 to 26.32 W m$^{-2}$ (28.61 to 29.15 W m$^{-2}$), and more than 80% of the 212 flux towers had a correlation greater than 0.70. As for the RMSE, 85% and 89% of the daily λE and H were less than 30 W m$^{-2}$, respectively (Supplementary Fig. S8). It was obvious that flux towers with large biases were mainly located on the coast of Australia and the west coast and Great Lakes region of the United States, as well as the Mediterranean region, all of which are strongly impacted by advection from neighboring open water bodies. The biases of the monthly λE (H) were smaller than the biases of the daily λE (H). More than 89% and 90% of the sites had an R greater than 0.70 (Supplementary Fig. S9). Meanwhile, the λE estimation at more than 88% of the sites and the H estimation at more than 89% of the sites showed an RMSE less than 30 W m$^{-2}$. Finally, the daily λE and H of each weather station over the past few decades were predicted by the well-trained ANN model. The spatial distribution patterns of mean annual λE and H are consistent with the results in the Fluxnet-MTE (Jung et al., 2011) (Supplementary Fig. S10). The Fluxnet-MTE is a mature and widely applied machine learning product that can be used as a bench work. This ensemble of statistical estimates of λE were obtained from the Department of Biogeochemical Integration (BGI) of the Max Planck Institute (MPI) (https://www.bgc-jena.mpg.de/geodb/projects/Data.php). The mean annual ET of the MET model ranged from 0 to 1400 mm (Jung et al., 2010), while the mean annual ET of this study ranged from 0 to 1416 mm during the 1982−2008 period (Supplementary Fig. S11). In different large-scale latitude intervals, the temporal changes of the ANN model estimated λE and the temporal changes of the MET model estimated λE are significantly correlated at the p<0.05 level, which further emphasizes the reliability of the ANN model retrieval results (Supplementary Fig. S12).

## 3.2 Attribution of trends in climate variables

The attribution of trends in climate variables were estimated for two reasons: (1) to quantify the changes in the atmospheric water supply, and (2) to estimate the long-term trends in atmospheric evaporative demand factors including VPD, air temperature, and surface wind speed. Aannual precipitation exhibited an increasing trend ranging from 3 to 40 mm per decade in western Europe, the United States, Southeast Asia, and Australia. Conversely, annual precipitation exhibited a decreasing trend ranging from -3 to -30 mm per decade in northern Eurasia, the savanna region of Brazil, and South Africa (Fig. 3a). In particular, annual precipitation showed a more obvious upward trend than before in a large area of land in recent period, i.e., 2001–2017 (Supplementary Fig. S13). Rising air temperature and the associated increasing water holding capacity of the atmosphere were the primary causes for the substantial increase in precipitation (Byrne et al., 2015), except for some regions (e.g., Russia) with insufficient moisture advection from ocean or regional evaporation.

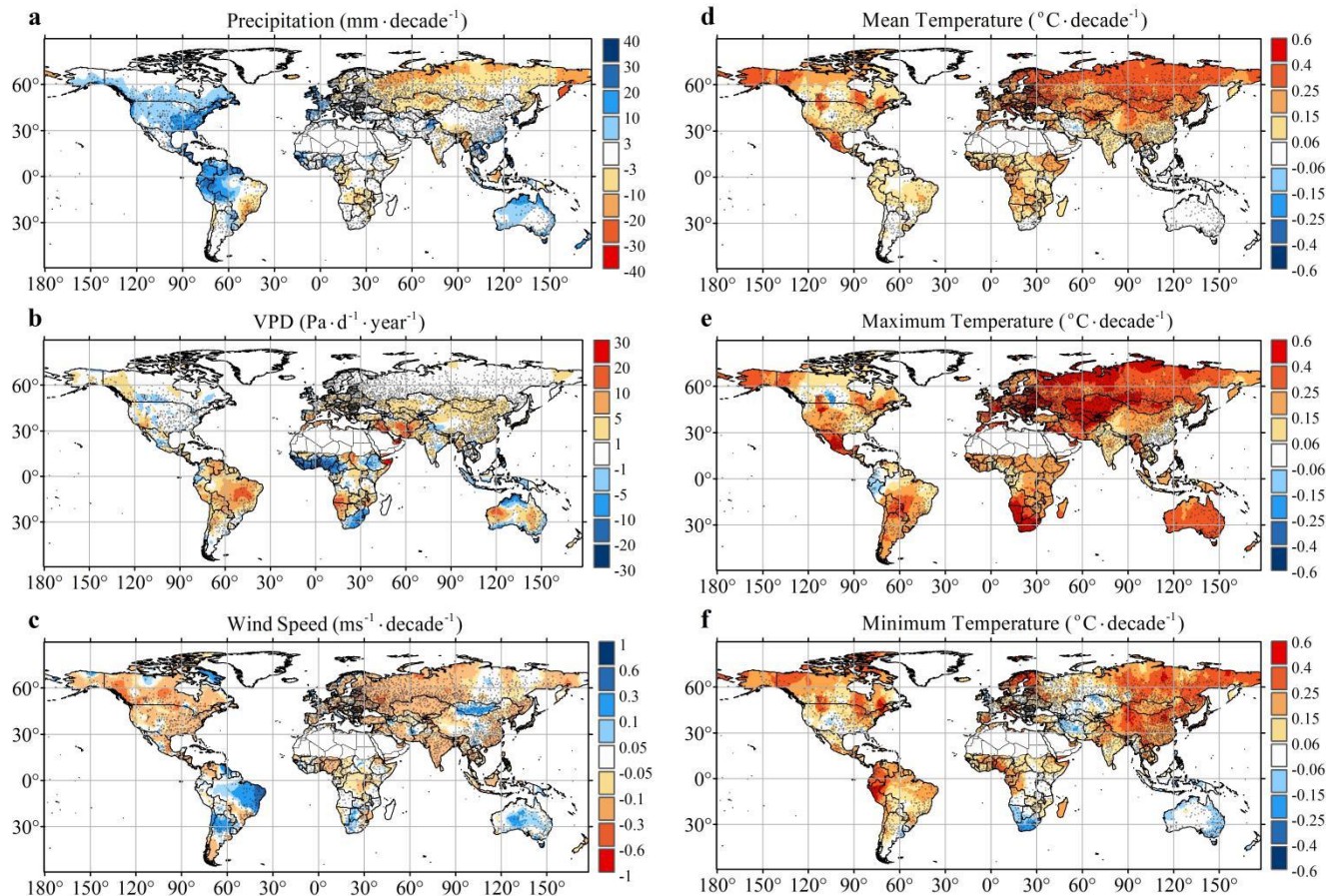

**Figure 3.** Long-term trends in annual precipitation, vapor pressure deficit (VPD), surface wind speed, mean temperature, maximum temperature, and minimum temperature. Values are not shown for the Greenland and Sahara region as there are scarce weather stations, and the range of the Sahara is referred to the existing study (Vicente-Serrano et al., 2015). Small gray squares show locations of the weather stations used to interpolate global patterns.

With respect to the atmospheric water demand sides, VPD primarily presented an increasing trend because of an increase in air temperature and a decrease in relative humidity, especially in the subtropics (Fig. 3b); this was consistent with the expectations of atmospheric dynamics and the influence of free-tropospheric warming (Held and Soden, 2006). Additional meteorological variables influencing the evaporative demand, such as the mean, maximum, and minimum temperatures, mostly presented increasing trends on the global scale, with the exception of a few areas, such as the United States/Canadian

Corn Belt, and Mexico, which showed signs of cooling due to agricultural irrigation (Thiery et al., 2017) (Fig. 3d−f). Therefore, both rising air temperatures and increased VPD indicate that the driving forces of soil evaporation and plant transpiration are increasing under the climate warming trend. In addition, mean surface wind speed—a meteorologic factor

associated with evaporation—showed an overall decreasing trend (i.e., global stilling) except in the Amazon, Argentina, Australia, and Mongolia (Fig. 3c).

**3.3 Long-term trends in EF, ET, and P–ET**

Annual EF ranges from 0 (full aridity stress) to 1 (no aridity stress), and it is an indicator of surface aridity linked to soil moisture availability and vegetation phenology, as well as the physiological effects of atmospheric $CO_2$ concentrations on vegetation (Francesco et al., 2014; Lemordant et al., 2018; Swann et al., 2016). The decreasing trend in the EF varied from 0 to 0.05 per decade and was prevalent in several land areas (Fig. 4a), except in the most humid areas of tropical rainforest

(e.g., the Amazon, West Africa, New Guinea Island, and Southeast Asia) and dense agricultural irrigation areas, including central North America and Punjab in India (Supplementary Fig. S14). Changes in the EF at different latitudinal intervals were consistent with the "dry gets drier, wet gets wetter" paradigm in the tropical areas (Chou et al., 2009; Liu et al., 2013). Moreover, the observed increase in EF further suggested a wet trend in western Sahel, where increasing rainfall was reported recently (Biasutti, 2019; Dong et al., 2015). It was systematically determined that the EF declined across large swaths of the

globe and exhibited different spatial patterns in different periods of the past few decades, which emphasized that this is not a short-term phenomenon (Fig. 5a–c). As the climate has warmed, decline in EF reflected an increase in surface resistance (see Methodology), which can be controlled by one of two factors—either an increase in stomatal resistance associated with the physiological effects of $CO_2$ or a decrease in soil moisture. Therefore, if soil moisture or surface runoff increases while EF decreases, it is a sign of increased surface resistance impacting the water balance.


The evolution of El Niño–Southern Oscillation (ENSO) can greatly influence global hydrological cycle and the patterns of aridity/wetness (Fu et al., 2012; Miralles et al., 2013; Nalley et al., 2019), and thus we analyzed the patterns of EF in different ENSO phases based on the multivariate ENSO index (MEI). However, no significant changes in EF trends were detected between different ENSO phases, with the exception of La Niña showing a significant impact on the aridity in East

Asia (Fig. 5d–f). In addition, the predicted EF trends using an ensemble from Phase 5 of the Coupled Model Intercomparison Project (CMIP5) under the Representative Concentration Pathway (RCP) 8.5 scenario (the warming scenario with the highest $CO_2$ emissions) also presented a decreasing trend in most global land areas (Supplementary Fig. S15a). Although the trend magnitudes vary across different periods, it indicated the direction of EF decline may be a long-term existing phenomenon. The model simulations further suggested that increasing $CO_2$ concentrations can affect the allocation of

surface energy and may cause a decrease in EF on large land surfaces.

As the climate warmed, ET showed a significant upward trend ranging from 0 to 0.03 mm per day per year (Fig. 4b), especially in the core regions of tropical rainforest climate zones (e.g., the Amazon, West Africa, and Southeast Asia), the coast of Australia, and the areas with a high density of agricultural irrigation (e.g., northern India, Central Asia, and Central

America). The increase in ET was primarily induced by the radiative effect of a warming climate, which can compensate for

the observed decrease trend in EF (ET=EF×$R_n$). Moreover, the observation-driven results showed a declining trend in ET at a rate of 0 to -0.03 mm per day per year on fractional land surfaces, such as North America, South Africa, Australia, Southeast Asia, and the Mediterranean region, which was consistent with the ET declining trends simulated by the CMIP5 climate models at RCP8.5 scenario (Supplementary Fig. S15b).

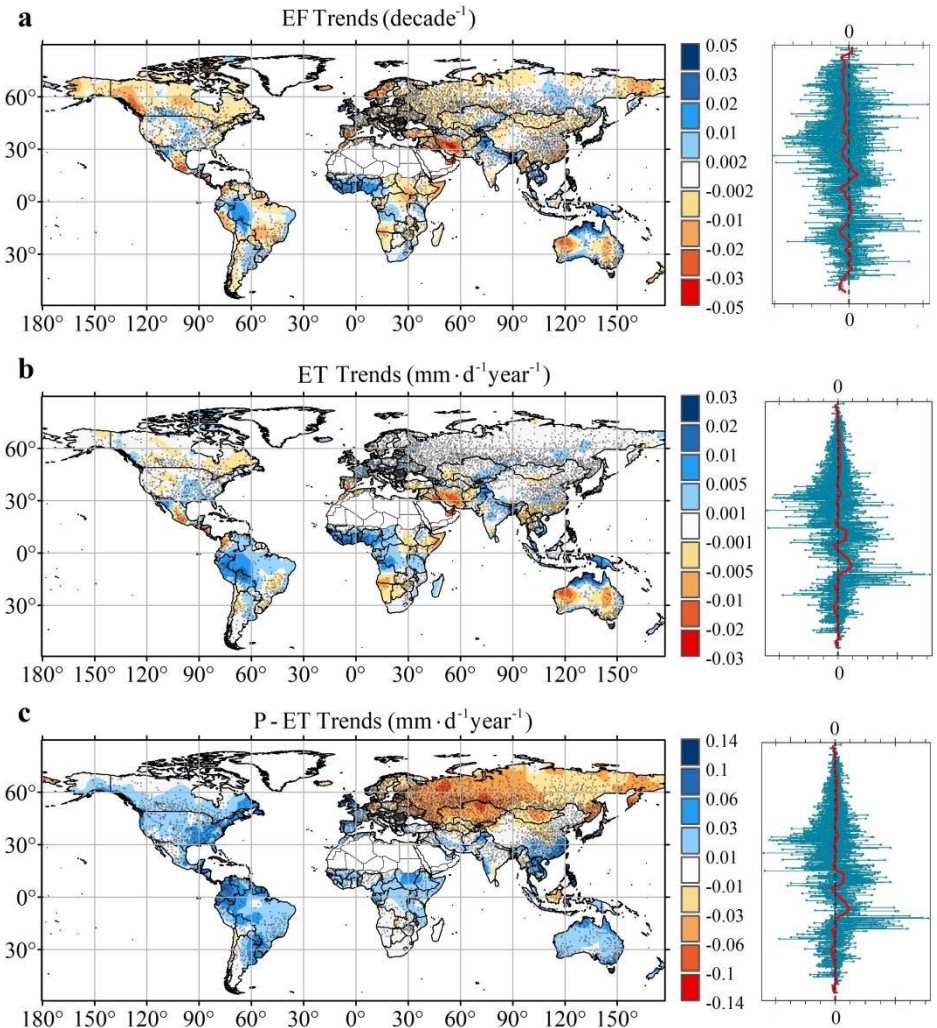

**Figure 4.** Long-term trends in evaporative fraction (EF), evapotranspiration (ET), and precipitation (P) minus ET (P−ET). ET was converted from the ANN retrieved latent heat flux. The red curve represents median trends at different latitudes.

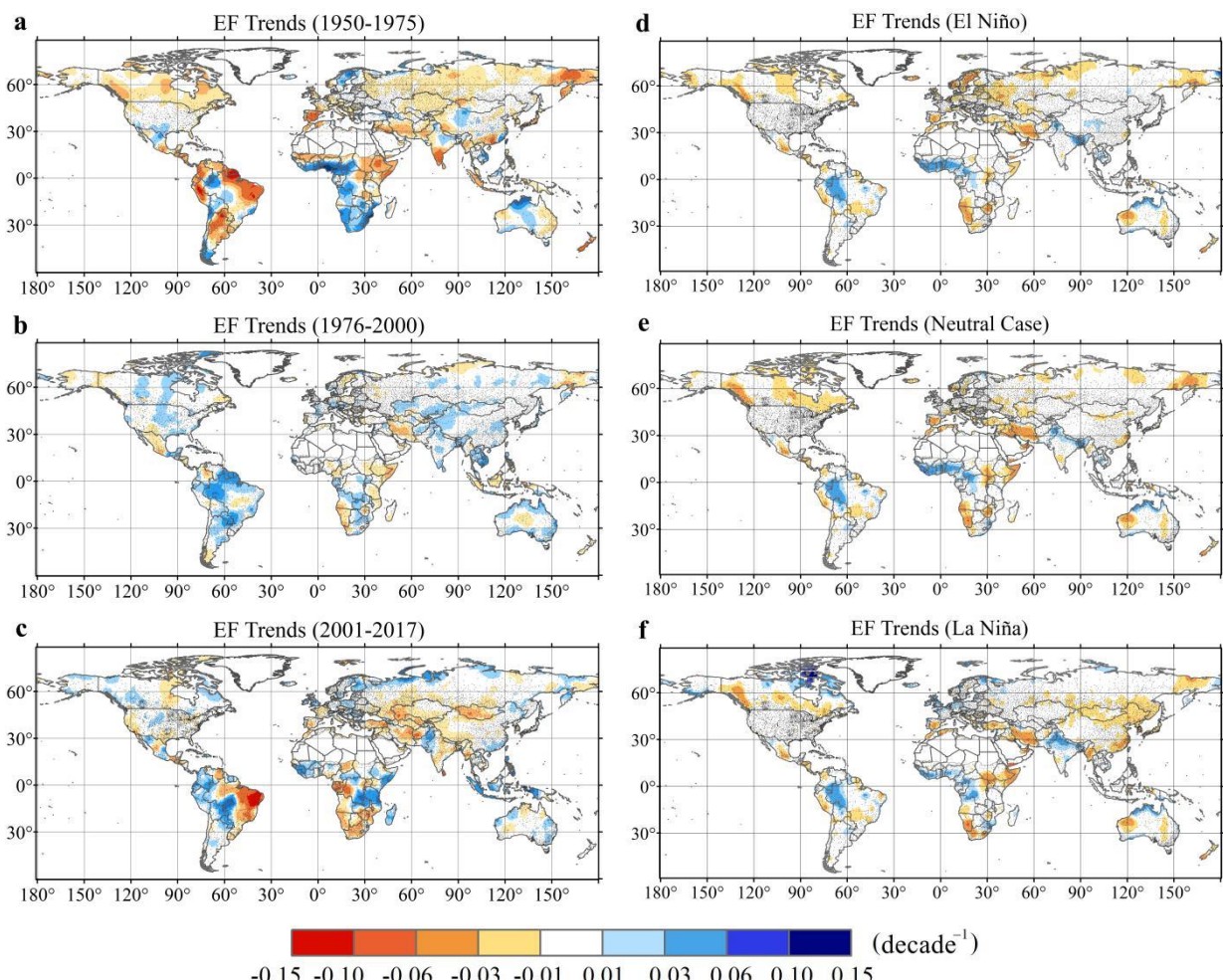

**Figure 5.** Spatial patterns of EF trends during different periods. (a–c) The spatial patterns show EF trends during different historical periods, and (d–f) the spatial patterns show EF trends during El Niño period, a neutral case period, and La Niña period, respectively.

P−ET, a proxy for long-term runoff, assumes that changes in storage due to human activity are negligible and are closely linked to water availability and soil moisture trends (Alkama et al., 2013; Sophocleous et al., 2002). Therefore, long-term runoff mainly presented an increasing trend on most of the global land, with the exception of a decrease in northern Eurasia (Fig. 4c). To verify the retrieved P−ET trend, we made an comparison between the P−ET trend and the observed runoff trend during the same periods in small- and medium-sized watersheds (5~1000 km$^2$) (Supplementary Fig. S16). The P−ET and observed runoff presented different trends in eastern Australia, which can be attributed to a decrease in runoff caused by human activities such as reservoir scheduling and agriculture irrigation (Bosmans et al., 2017; Lehner et al., 2011). When we only considered the stations that are not too influenced by large reservoirs, we found that the direction and the spatial pattern

of P–ET trend (Fig. 4c) are more obvious consistent with observed runoff trend, including the upward trend in northern Australia and the downward trend in southern Australia, the upward trend in western Europe and the downward trend in eastern Europe (Supplementary Fig. S16c). The spatial pattern of P–ET trends and observed runoff trends are also generally consistent in other regions including North and South America, Southern Africa, East Asia, and Southeast Asia. We do not fully expect the P-ET to be completely consistent with observed streamflow, because in addition to measurement errors, the
streamflow is strongly affected by human activities especially over long-term period. Model predictions also showed an overall increasing trend in P−ET (Supplementary Fig. S15c), while a decrease was predicted (but it has not been observed) in western United States and western Europe and P−ET was predicted to increase in northern Eurasia.

### 3.4 Signs of covariations in long-term EF and runoff

The signs of covariations in normalized ET, i.e., EF, and normalized P−ET, i.e., 1−ET/P, were further investigated to determine the patterns of surface aridity. We superimposed the EF trend, indicative of changes in aridity stress (e.g., temperature and soil moisture) or plant physiological effects (see Methodology), and the 1−ET/P trend, which was indicative of changes in long-term runoff. Land areas with a decreased EF and an increased 1−ET/P were indicative of a dominance of $CO_2$ plant physiological effects, because a long-term decline in ET with increasing runoff mainly attributable to surface
vegetation control. A decline in the EF caused by a decrease in surface conductance can be offset by an increase in the EF caused by the effects of climate warming. Nevertheless, in 27.06% of the global land areas, the EF has declined and has been accompanied by an increase in long-term runoff, which has been observed in most of North America, parts of South America, the Mediterranean, Africa, Australia, and Southeast China (Fig. 6). These signals further emphasized that surface vegetation controls and its response to changing environment have a great influence on water cycle and surface aridity variability.

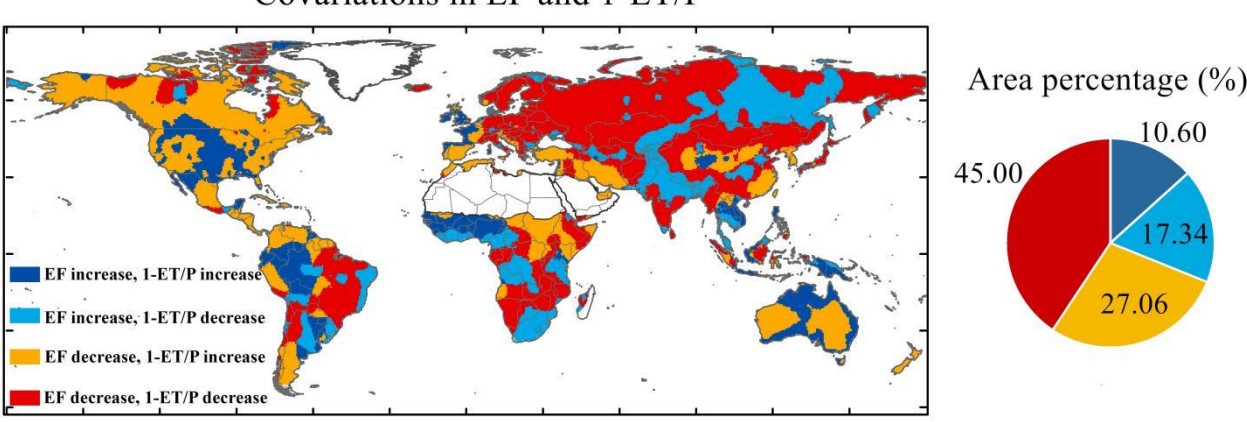


**Figure 6.** Signs of covariation in EF and 1–ET/P. Right panel shows area percentage of different signs, and the area fractions are calculated by the spherical area.

In addition, the signs of increase in EF with decreasing runoff accounted for 17.34% of the global land areas, which was mainly due to agricultural irrigation and land use management, such as in the Punjab region of India, Central Asia, and downstream Amazon where there is a high density of irrigation (Supplementary Fig. S14). The land areas showing increase trend in both EF and runoff were typically located in humid regions and accounted for 10.60% of the global land surface. With the increase of EF and 1–ET/P, the humid areas of the Amazon, West Africa, Southeast Asia, and the coast of Australia are getting wetter (Fig. 6). Particularly, the previously reported wet trend in western Sahel was captured by the increase trends in both EF and 1–ET/P. Additionally, 45.00% of the global land areas experienced a decreasing trend in EF and 1−ET/P, and thus aridity stress posed a relatively larger risk to these regions. EF and 1–ET/P both exhibited a decreasing trend in the arid regions of the Amazon (e.g., the savanna region of Brazil), and thus those areas are getting drier. Moreover, the Mediterranean region, northern Eurasia, and South Africa also experienced a decrease trend in EF and 1–ET/P, which was consistent with the existing observation analysis and predictions (Padrón et al., 2020; Samaniego et al., 2018; Zhou et al., 2019).

## 4 Concluding Remarks

This study for the first time provided the strategy for retrieving consistent latent heat and sensible heat fluxes on a global scale, based on boundary layer energy budget perspective and machine learning approach driven by the ground observations of globally distributed flux towers and weather stations. After that, we quantified the attributions of long-term changes in surface aridity/wetness. The latent heat and sensible heat fluxes retrieved in this study can be an important supplement to the existing product. Our study have important implications for understanding variability of surface aridity under changing environment and providing constraints for model predictions. Although we attempted to infer surface energy fluxes from ground observations and used various data quality control methods to reduce uncertainty, the quality of the observational data from flux towers and weather stations can influence our retrievals.

In the absence of surface regulation of plant physiological effect and changes in biomass, a warming climate was expected to intensify ET at a rate roughly governed by the Clausius–Clapeyron relation. However, a long-term relative decrease in normalized ET accompanied by increasing runoff was found in 27.06% of the global land areas, which was indicative of a reduction in surface conductance. The findings further emphasized that vegetation controls have strong impacts in regulating the water cycle and surface aridity variability. Climate models have captured some of these changes; however, they have also exhibited large regional discrepancies. Therefore, representations of land use management and plant physiological effects are essential for the improvement of future predictions with respect to water, energy, and carbon cycles.

*Data/code availability.* The data, results and Matlab codes in this study are available upon request. The eddy-covariance data are available at http://fluxnet.fluxdata.org/. The Global Summary of the Day and the Global Historical Climatology Network datasets are collected from the NOAA at https://www.ncdc.noaa.gov/data-access. The data of Global Runoff Data Center are available at https://www.bafg.de/GRDC/EN/01_GRDC/13_dtbse/database_node.html. The reservoirs data in the GRanD database are available at http://sedac.ciesin.columbia.edu/pfs/grand.html. Global irrigation data are available at http://www.fao.org/nr/water/aquastat/irrigationmap/index10.stm. The Multivariate ENSO Index (MEI) is available at https://www.esrl.noaa.gov/psd/enso/mei/.

*Author contributions.* RW and PG designed the study. RW performed the experiment, analyzed the results, and wrote the manuscript. PG designed the methodology, analyzed the results, and revised the manuscript. JY and LC contributed to data collection and validation. JC and LL contributed to discussion and supervision.

*Competing interests.* The authors declare that they have no conflicts of interest.

*Acknowledgments.* We acknowledge the members of FLUXNET community for sharing flux tower observational data, and Ren Wang acknowledge Dr. Léo Lemordant for helping with the Earth System Models simulation. We also thank Dr. René Orth and the other anonymous reviewer for providing valuable comments.

*Financial support.* This work was financially supported by the National Key Research and Development Program of China (2017YFA0603603), the China Postdoctoral Science Foundation (2020M681656) and the China Scholarship Council scholarship (201706380063). JB Yin acknowledge the National Natural Science Foundation of China (52009091).

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
