# Peer review of "Long-term relative decline in evapotranspiration with increasing runoff on fractional land surfaces"

_Hydrology and Earth System Sciences, 2020_

## Referee Comment (RC1) · Rene Orth (Referee) · 16 Dec 2020

Review of Wang et al. "Long-term relative decline in evapotranspiration with increasing runoff on fractional land surfaces"

This study derives land evapotranspiration with a machine learning approach applied to sub-daily meteorological station data from across the world. More decreasing than increasing trends are found across the Earth's land areas. The controls for these trends are determined and distinguished by jointly considering trends in evapotranspiration and precipitation minus evapotranspiration as a proxy for runoff.

[Figure]

——————-

Recommendation: I think the paper requires major revisions.

The methodology and research question addressed in this paper are novel and relevant, making it a potentially good fit for HESS. Also the joint consideration of evapotranspiration and (proxy) runoff to interpret the reported trends and associate them with potential causes is an important contribution to the land surface science community. However, before the paper is suitable for publication in HESS, some critical shortcomings need to be addressed:

(1) I like that the authors validated their simulation results against observed streamflow and a reference ET product. However, the diagnosed agreement with these products is actually not very convincing, particularly in terms of the trends, as shown in Figure S8, and in the comparison between Figures 4c and S11. I think it is critical to understand these differences between the products, as otherwise I am missing convincing evidence that the data simulated here can be used to assess global ET and runoff trends.

(2) Adding to (1), it would be insightful to expand the cross-validation analysis from Figure 2 to validate the derived data also in terms of the trends observed at the independent cross-validation stations, as the final conclusions of this study are build on the trends rather than the short-term variability of the data.

(3) The comparison of ET and runoff trends between the CMIP5 scenario simuations and the machine learning-derived, historical simulations, does not really make sense as totally different time periods are considered to compute the respective trends, if I get this right?

(4) The description of the employed machine learning algorithm is not clear. The choice of artificial neural networks over other machine learning methods is not sufficiently motivated. Why is it more suitable than for example random forests in this context? Why

not simply use FLUXCOM instead of deriving yet another estimate? Further, the setup of the ANN model is unclear, i.e. how the hyper-parameters are chosen (why exactly 2 hidden layers? why 500 epochs?), why different performance metrics (RMSE and MSE) are chosen, how the training is done, and how overfitting is prevented. I acknowledge that some of these choices are necessarily arbitrary, but this would be good to mention, including some tests on the relevance of these choices for the conclusions of the study.

(5) There are many small language errors (such as missing articles or wrong grammar) throughout the manuscript. The authors need to take special care of these when revising the manuscript.

I do not wish to remain anonymous - René Orth.

—————————

Specific comments:

lines 33 & 36: Please explain what you mean with "offline"

lines 57, 72, 83, 123/124: You mention in these places different sets of variables which are (not?) used by the ANN algorithm, please clarify

line 69: How is this gap filling done? can it affect your results?

line 71: Why are you targeting daily resolution? To infer trends, monthly resolution might be sufficient?

lines 86-91: I do not fully understand this paragraph. Do the original and target station groups differ by the number of stations in some grid cells? If yes, do you average across multiple stations in one grid cell, or remove stations to only keep one (and which one?)?

line 101: Why do you emply top-of-atmosphere radiation instead of surface radiation which is what the vegetation is actually exposed to?

lines 156/157: Where is the influence of the resistances shown, or how do you arrive to this conclusion?

line 160: How was the cross validation done? How/which stations or times have been chosen as independent validation data?

line 168: I would think that also SAV is water-limited?

lines 165, 169: Why are there different correlations given for OSH?

line 184/185: What does 0.78∼0.79 and 0.77∼0.78 mean?

line 195: The reference for FLUXCOM would be Jung et al. 2019 which is also in the reference list, or did you actually use the MTE product from Jung et al. 2010?

line 220: I cannot see a cooling trend in northern Europe in Figure 3.

lines 226-228: Phrasing could be improved, the "when" is not needed here.

line 235: Not sure if I would see a "persistent long-term trend" anywhere here as the spatial patterns of trends vary quite a lot across time periods in Figure 5.

line 245: Which models did you consider in particular?

line 261: There is no "RCP8.5 climate model".

section 3.4: Very nice approach and analysis to infer potential causes of the observed trends.

Figure 1: What are the black empty bars which are superimposed on the colored bars?

Figure 3: How is the Sahara desert defined? Why are other deserts not excluded, too? What is the time period over which the trends are computed? How is the spatial interpolation between the station locations done?

Figure S10: This is cool, but where is the information coming from?

Tables S1 and S2: Some of the variable names here need more explanation. And why

was only the u-component of the wind speed used?

References: Jung, M., et al. The FLUXCOM ensemble of global land-atmosphere energy fluxes, Scientific Data 6, 74 (2019). Jung, M., et al. Recent decline in the global land evapotranspiration trend due to limited moisture supply. Nature 467, 951–954 (2010).

---

## Referee Comment (RC2) · Anonymous Referee #2 · 3 Jan 2021

The present study investigated the ratio of latent heat flux to available surface energy (EF) using an ANN method and FLUXNET and meteorological station data, and reported that EF decreased on a fractional land surface, especially, it was accompanied by increased runoff (precip – et). The topic of the study falls into the scope of the HESS journal, and the conclusion is interesting. A minor revision is recommended before its publication.

Major concerns: a) The validation of the ANN method needs further clarifications. From lines 161- 162, r ranged from 0.782 to 0.768, corresponding to a R2 of 0.59-0.61. More clarifications are need to prove that such accuracy is acceptable. Probably, the

authors could compare the accuracy obtained in the study with those in previous similar studies. b) The writing of introduction and conclusion sections need to be improved. The introduction section: this section mainly stated that the traditional method did not consider the dynamic change of leaf stomatal resistance/conductance, while info about the similar studied based on observed data is a little bit limited. The conclusion section: this section is too simple, only the sentence starting with 'however' is a conclusion. Please add more info to this section.

Minor concerns: Line 18: 'namely that...' should be changed to 'namely, ...' Lines 35-38: the description about the ET output from models is incorrect, cause most simulated ET output actually has already considered insufficient soil moisture's influences. Line 64: 'EF' seems to appear for the first time here, so full name is needed for EF. Line 72: radiation is missing after 'shortwave'? Line 82: Station meteorological data might be a better caption. Line 135: 'rs' appears for the first time here? If so, the full name is needed. Line 242: References of Fu et al. (2012WRR, 2015JGR-A) are recommended here to illustrate ENSO's influences.

---

## Author Comment (AC1) · 27 Jan 2021

**Response to Referee #2**

**Comment (1):** The present study investigated the ratio of latent heat flux to available surface energy (EF) using an ANN method and FLUXNET and meteorological station data, and reported that EF decreased on a fractional land surface, especially, it was accompanied by increased runoff (precip–et). The topic of the study falls into the scope of the HESS journal, and the conclusion is interesting. A minor revision is recommended before its publication.

**Reply:** We thank the reviewer's valuable suggestions and constructive comments, which help us improving our study and the quality of the manuscript. The comments and suggestions are addressed below.

**Comment (2):** Major concerns: a) The validation of the ANN method needs further clarifications. From lines 161-162, r ranged from 0.782 to 0.768, corresponding to a $R^2$ of 0.59-0.61. More clarifications are need to prove that such accuracy is acceptable. Probably, the authors could compare the accuracy obtained in the study with those in previous similar studies.

**Reply:** Thanks for the valuable suggestion. We have expanded the cross-validation in terms of not only values but also trends and found that the trends predicted by the ANN model were highly correlated with the observed trend, and even in most cases, the estimation of the trends were more reliable than the estimation of the values (see the Attached Figure at the end). In some land cover types such as ENF, DBF, GRA, and WET, the statistical correlations in the trends validation exceed 0.90 (p<0.001), and those statistics are the ones being reported. Meanwhile, we will provide more discussion here through comparing the accuracy with previous similar studies according to the comment.

**Comment (3):** b) The writing of introduction and conclusion sections need to be improved. The introduction section: this section mainly stated that the traditional method did not consider the dynamic change of leaf stomatal resistance/conductance, while info about the similar studied based on observed

data is a little bit limited. The conclusion section: this section is too simple, only the sentence starting with 'however' is a conclusion. Please add more info to this section.

**Reply:** Following the comment, we will extend the introduction and conclusion, including on the studies of using observation-driven latent heat and sensible heat fluxes in the introduction section, and we will improve and refine the conclusions of this study.

**Comment (4):** Minor concerns: Line 18: 'namely that. . .' should be changed to 'namely, . . .'

**Reply:** After considering the suggestion, we have modified 'namely that...' to 'that is,' in the revised manuscript.

**Comment (5):** Lines 35-38: the description about the ET output from models is incorrect, cause most simulated ET output actually has already considered insufficient soil moisture's influences.

**Reply:** We agree with this comment, and we have modified the expression of this sentence. We want to emphasize that some traditional drought assessments use the output of climate models (e.g., predicted temperature) to estimate potential ET. This is an offline method that can ignore insufficient soil moisture's influences.

**Comment (6):** Line 64: 'EF' seems to appear for the first time here, so full name is needed for EF.

**Reply:** Done.

**Comment (7):** Line 72: radiation is missing after "shortwave"?

**Reply:** We have added the term "shortwave" in the revised manuscript.

**Comment (8):** Line 82: Station meteorological data might be a better caption.

**Reply:** According to the comment, we have modified the caption to "Observed weather station data", which is consistent with the expression in the full text.

**Comment (9):** Line 135: 'rs' appears for the first time here? If so, the full name is needed.

**Reply:** We have added the full name and avoid such omissions in the revised manuscript.

**Comment (10):** Line 242: References of Fu et al. (2012WRR, 2015JGR-A) are recommended here to illustrate ENSO's influences.

**Reply:** Thank you very much for your recommendations. After studying the papers, we find that Fu et al (2012) is very suitable to illustrate the influences of ENSO, and we will add this paper as one of the references in the revised manuscript.

**References:**

Fu, C., James, A. L., and Wachowiak, M. P. Analyzing the combined influence of solar activity and El Niño on streamflow across southern Canada. Water Resour. Res., 48, W05507. https://doi.org/10.1029/2011WR011507, 2012.

[Figure]

**Figure 2.** Density scatter plot for (a) the cross-validation in terms of values and (b) the cross-validation in terms of trends. Samples of the validation set in the values cross-validation are randomly composed of 10 flux towers from different plant function types, and the validation set in the trends cross-validation are composed of trends calculated from all time periods.

---

## Author Comment (AC2) · 27 Jan 2021

**Response to the reviewers' comments**

**Response to Referee #1**

We greatly appreciate Dr. René Orth providing valuable and constructive comments on our manuscript HESS-2020-590. We seriously considered each comment and revised the manuscript accordingly. The individual comments are replied below. The comments are shown in black font and our responses are shown in blue font.

**Comment (1):** This study derives land evapotranspiration with a machine learning approach applied to daily meteorological station data from across the world. More decreasing than increasing trends are found across the Earth's land areas. The controls for these trends are determined and distinguished by jointly considering trends in evapotranspiration and precipitation minus evapotranspiration as a proxy for runoff.
* * *
Recommendation: I think the paper requires major revisions.

The methodology and research question addressed in this paper are novel and relevant, making it a potentially good fit for HESS. Also the joint consideration of evapotranspiration and (proxy) runoff to interpret the reported trends and associate them with potential causes is an important contribution to the land surface science community. However, before the paper is suitable for publication in HESS, some critical shortcomings need to be addressed.

**Reply:** Thanks for your encouraging and constructive comments. All your comments are addressed in the revised manuscript, and we hope you will find the latest version suitable for publication.

**Comment (2):** I like that the authors validated their simulation results against observed streamflow and a reference ET product. However, the diagnosed agreement with these products is actually not very convincing, particularly in terms of the trends, as shown in Figure S8, and in the comparison between Figures 4c and S11. I think it is critical to understand these differences between the products, as otherwise I am missing convincing evidence that the data simulated here can be used to assess global ET and runoff trends.

**Reply:** We agree with the point that the differences between the products should be better understood. The difference between our ET and the ET estimated by model tree ensemble (MTE) can be caused by the different models, driven data, and time scales. Meanwhile, we should not fully expect the P-ET to be completely consistent with observed streamflow, because (1) there are inherent differences between atmospheric scale and hydrologic scale, (2) the conversion of P-ET to streamflow largely depends on the

underlying surface, and (3) the observed streamflow is strongly affected by human activities especially on long time scales. We also note that because our work is based on a boundary layer perspective where physics does not change in time, whereas direct estimates of ET are affected by additional things like nutrients or increased $CO_2$ whose trends cannot be captured by the Fluxnet data.

According to the comment, we added more discussions and expanded the cross-validation in terms of trends as suggested in the comment (3).

**Comment (3):** Adding to (1), it would be insightful to expand the cross-validation analysis from Figure 2 to validate the derived data also in terms of the trends observed at the independent cross-validation stations, as the final conclusions of this study are build on the trends rather than the short-term variability of the data.

**Reply:** Thanks for the suggestion. We expanded the cross-validation analysis in terms of trends in the revised manuscript. It was suggested that the trends predicted by the ANN model were highly correlated with the observed trends, and even in most cases, the estimation of the trends were more reliable than the estimation of the values (please see the Attached Figures at the end).

**Comment (4):** The comparison of ET and runoff trends between the CMIP5 scenario simulations and the machine learning-derived, historical simulations, does not really make sense as totally different time periods are considered to compute the respective trends, if I get this right?

**Reply:** Yes, the trends vary from different time periods. We acknowledge that a consistent time period is better, but it is difficult to achieve due to paucity of data. Moreover, the main purpose here is to identify the change directions of ET or runoff rather than their magnitudes. Thus, we can determine whether the trends of decrease (increase) in ET or runoff is a long-term existing phenomenon by comparing the change directions over different time periods in history and in simulated future scenarios.

After considering the comment, we will add an explanation in section 3.3 in the revised manuscript.

**Comment (5):** The description of the employed machine learning algorithm is not clear. The choice of artificial neural networks over other machine learning methods is not sufficiently motivated. Why is it more suitable than for example random forests in this context?

**Reply:** Different machine learning algorithms have their own advantages. The artificial neural networks tend to work well, as it has strong nonlinear function approximation capability and fault tolerance. As our selected neural networks demonstrated good performance, to save space, we did not compare different machine learning algorithms in this study.

We still fully considered your comments and added a description on the motivation of using neural

networks in section 2.4.

**Comment (6):** Why not simply use FLUXCOM instead of deriving yet another estimate?

**Reply:** The data in the FLUXCOM or the results estimated by the model tree ensemble (MET) are driven by monthly remote-sensing and meteorological reanalysis data, and thus they rely on the satellite era and cannot be used for long-term trends. In addition to the too short record required for correct trend estimates, evapotranspiration can be modified by subtle changes in $CO_2$ or nutrients, which can modify transpiration for instance or there can be even a decoupling between surface and deeper soil moisture (Berg et al., 2016), none of which are captured by FLUXCOM. This is why we use an opposite view – we use in essence a boundary layer budget based on Salvucci and Gentine (2015) and Gentine et al. (2016) except that we lump the non-linearity for the boundary layer budget in a neural network. The physics of the boundary layer is not changing in time and weather stations have been available for many decades, allowing the development of the first long-term record of hydrologic trends. Finally, FLUXNET-MTE and FLUXCOM are based on data that are highly localized, especially in the northern hemisphere. The network of weather station is both much longer but also extend to much more remote places such as in the tropics, providing much more constrain in those places, where other retrievals typically display very large uncertainties. The aim of our strategy in this study is therefore to infer a longer surface fluxes (as well as better generalization to the tropics and other remote regions). Thus, the length of the surface heat fluxes in existing products does not match with our research purpose. After considering this comment, we will add an explanation in introduction section.

**Comment (7):** Further, the setup of the ANN model is unclear, i.e. how the hyper-parameters are chosen (why exactly 2 hidden layers? why 500 epochs?). Why different performance metrics (RMSE and MSE) are chosen, how the training is done, and how overfitting is prevented. I acknowledge that some of these choices are necessarily arbitrary, but this would be good to mention, including some tests on the relevance of these choices for the conclusions of the study.

**Reply:** The more hidden layers and neurons in the ANN, the stronger the nonlinear ability of the model, but the complexity and training time are also increasing. In theory, a neural network with 2 hidden layers can realize any complex mapping, as the nonlinear ability can be enhanced by adding neurons. The ANN model with 2 hidden layers and 15 neurons shows good performance and appropriate time consumption (see the Attached Fig. S2). As for the optimal number of neurons, we initially determined it according to an empirical formula, i.e.,

$$h = \sqrt{(n+m)} + a$$

.

Where $n$ is the number of input neurons, and $m$ is the number of output neurons, and $a$ is a constant ranging from 0 to 10.

MSE was an indicator used to evaluate the performance of neural works in the training process of adjusting weight, and RMSE is an indicator used to analyze the bias between the ANN predicted values and the observed values in validation set.

To avoid over-fitting, the early stopping method was used to avoid overfitting this study, that is, we recorded the best validation accuracy during the training process, and the training was stopped when the MSE was no longer reduced after going through the entire dataset.

We apologize for the omission of some descriptions on the setup of training the ANN, and we have added more information on the process in the section 2.4 accordingly.

**Comment (8):** There are many small language errors (such as missing articles or wrong grammar) throughout the manuscript. The authors need to take special care of these when revising the manuscript.

**Reply:** The language has been polished by a language editing agency, and we will take special care for each sentence and add missing articles in the references.

**Comment (9):** lines 33 & 36: Please explain what you mean with "offline".

**Reply:** The calculation of potential evaporation in drought index using meteorological variables from climate model outputs is offline. We have modified this sentence to "Using potential evaporation rather than actual ET or calculating offline ET using meteorological variables from climate model outputs in the traditional drought indices, the calculation implicitly assumes that soil can always supply moisture to meet the atmospheric evaporation demand".

**Comment (10):** lines 57, 72, 83, 123/124: You mention in these places different sets of variables which are (not?) used by the ANN algorithm, please clarify.

**Reply:** Top-of-atmosphere shortwave radiation, relative humidity, temperatures (mean, maximum, and minimum temperatures), and surface wind speed are the variables used by the ANN algorithm (in line 123/124).

The expressions of temperatures, humidity, and solar radiation (in line 57) are a broad concept and does not refer to a specific variable. We have rephrased this sentence to "This approach utilizes daily observations of meteorological variables such as temperatures, humidity, and solar radiation."

The expressions of top-of-atmosphere shortwave radiation, vapor pressure deficit (VPD), mean temperature, and surface wind speed (in line 72) are referring to the data collected from the integrated daily product of FLUXNET2015. VPD is used to calculate relative humidity, and daily maximum and

minimum temperatures are obtained from half-hourly/hourly flux tower measurements. These data were used to train the ANN model.

The expressions of precipitation, temperatures (mean, maximum, and minimum temperatures), dew point temperature, and surface wind speed (in line 83) are referring to the data collected from weather stations. Dew point temperature was used to calculate relative humidity at weather stations, and the meteorological data collected from global weather stations were used to drive the trained ANN models.

According to the comment, we have modified the expressions to ensure that the information is clear in the revised manuscript.

**Comment (11):** line 69: How is this gap filling done?

**Reply:** The gap-filling data were provided by the FLUXNET.

**Comment (12):** line 71: Why are you targeting daily resolution? To infer trends, monthly resolution might be sufficient?

**Reply:** We agree with that monthly scale might be sufficient for inferring trend, but daily scale is a true time scale in reality and thus it can capture the daily-cycle signal of water and heat fluxes. There is no contradiction between retrieving surface fluxes on daily scale and on monthly scale, as monthly fluxes can be converted from daily results.

**Comment (13):** lines 86-91: I do not fully understand this paragraph. Do the original and target station groups differ by the number of stations in some grid cells? If yes, do you average across multiple stations in one grid cell, or remove stations to only keep one (and which one?)?

**Reply:** Thanks for the comment, and we have improved the writing of this paragraph. The target stations were obtained according to the following three steps, i.e., (1) The stations with a time series spanning less than ten years were excluded; (2) if the stations had the same geographic coordinates, we used the stations with longer observation to replace the stations with shorter observation; (3) if there were multiple stations showing different coordinates in a 0.1-degree grid, we removed the stations with a shorter observation length.

**Comment (14):** line 101: Why do you employ top-of-atmosphere radiation instead of surface radiation which is what the vegetation is actually exposed to?

**Reply:** Because there do not exist reliable long-term surface observational solar radiation data, top-of-atmosphere shortwave radiation is a good replacement. After considering this comment, we have added an explanation in section 2.3.

**Comment (15):** lines 156/157: Where is the influence of the resistances shown, or how do you arrive to this conclusion?

**Reply:** We demonstrate the influence in an indirect way by inferring changes from a big leaf model (which is not used to derive the ANN). According to this comment and the Eq. (14), we have modified the sentences as following:

"Therefore, a decline in EF is linked with surface resistance ($r_s$) and there is a negative relationship between $r_s$ and EF. Annual EF ranges from 0 to 1 and $r_a$ is a function of wind speed. Thus changes in $r_a$ are relatively small while changes in $r_s$ can be strong."

**Comment (16):** line 160: How was the cross validation done? How/which stations or times have been chosen as independent validation data?

**Reply:** We randomized the samples of five flux towers (moderate number) from 212 sites as the validation set, and then used the remaining samples to train the ANN model. The ANN predicted daily λE (H) of the validation set were compared with their observed values. As for validation under different land covers, we randomly select samples from five flux towers under this land type as the validation set.

According to this comment, we have added a description of the random selection of cross-validation samples. Meanwhile, we have redrawn the Fig. 2 by randomly selecting 10 flux towers from different plant function types as the validation set (see Fig. 2 in the Attached Figures).

**Comment (17):** line 168: I would think that also SAV is water-limited?

**Reply:** Agree and done.

**Comment (18):** lines 165, 169: Why are there different correlations given for OSH?

**Reply:** Apology for the typo. We have corrected it in the revised manuscript.

**Comment (19):** line 184/185: What does 0.78~0.79 and 0.77~0.78 mean?

**Reply:** The range of correlation coefficients. We have corrected these sentences.

**Comment (20):** line 195: The reference for FLUXCOM would be Jung et al. 2019 which is also in the reference list, or did you actually use the MTE product from Jung et al. 2010?

**Reply:** Thanks for the reminder. We used the ensemble data of latent heat flux on land from the Department of Biogeochemical Integration (BGI) of the Max Planck Institute (https://www.bgc-jena.mpg.de/geodb/projects/Data.php). The data were retrieved using the model tree

ensemble (MTE) approach for upscaling FLUXNET measurements (Jung et al., 2011). We have corrected the reference and relevant expressions in the revised manuscript.

**Comment (21):** line 220: I cannot see a cooling trend in northern Europe in Figure 3.

**Reply:** We have corrected it in the revised manuscript.

**Comment (22):** lines 226-228: Phrasing could be improved, the "when" is not needed here.

**Reply:** We have rephrased this sentence.

**Comment (23):** line 235: Not sure if I would see a "persistent long-term trend" anywhere here as the spatial patterns of trends vary quite a lot across time periods in Figure 5.

**Reply:** We have revised the sentence to avoid inaccurate expression.

**Comment (24):** Which models did you consider in particular?

**Reply:** The simulation is an ensemble from Phase 5 of the Coupled Model Intercomparison Project (CMIP5) under the RCP8.5 scenario. We have modified the relevant expression.

**Comment (25):** line 261: There is no "RCP8.5 climate model".

**Reply:** Apology for the wrong expression, and we have corrected it accordingly.

**Comment (26):** section 3.4: Very nice approach and analysis to infer potential causes of the observed trends.

**Reply:** Thanks for the favorable evaluation.

**Comment (27):** Figure 1: What are the black empty bars which are superimposed on the colored bars?

**Reply:** Black empty bars represent the towers in the FLUXNET2015 Dataset and the solid bars represent all towers registered in FLUXNET.

We have redrawn the Figure 1 and its related Figure S1a to make the illustration clear (see Fig. 1 and Fig. S1 in the Attached Figures).

**Comment (28):** Figure 3: How is the Sahara desert defined? Why are other deserts not excluded, too? What is the time period over which the trends are computed? How is the spatial interpolation between the station locations done?

**Reply:** There is no strict definition of the scope of the Sahara desert. We prefer excluding the Sahara

because it has the largest desert area and scarce meteorological observation data. We referred to previous drought studies such as Vicente-Serrano et al., 2015, which also did not consider the Sahara region.

The trends are computed over the period of 1950-2017. If the observational data is not that long, the trends are converted to a uniform length of 68 years. The spatial interpolation in this study uses the Kriging interpolation method based on ArcGIS platform.

**Comment (29):** Figure S10: This is cool, but where is the information coming from?

**Reply:** Data are collected from the Food and Agriculture Organization of the United Nations (http://www.fao.org/nr/water/aquastat/irrigationmap/index10.stm). We have added the information of data source in the revised manuscript.

**Comment (30):** Tables S1 and S2: Some of the variable names here need more explanation. And why was only the u-component of the wind speed used?

**Reply:** We have provided more explanations about the variable names in Tables S1 and S2 (see Attached Tables), and have improved the writing of this section.

The wind speed used in this study is mean surface wind speed of the day, not the u-component of the wind speed. We have modified the abbreviation of wind speed.

**Some papers are added in the References, i.e.,**

Jung, M., Reichstein, M., Margolis, H. A., Cescatti, A., Richardson, A. D., Arain, M. A., Arneth, A., Bernhofer, C., Bonal, D., Chen, J., Gianelle, D., Gobron, N., Kiely, G., Kutsch, W., Lasslop, G., Law, B. E., Lindroth, A., Merbold, L., Montagnani, L., Moors, E. J., Pagpale, D., Sottocornola, M., Vaccari, F., and Williams, C.: Global patterns of land-atmosphere fluxes of carbon dioxide, latent heat, and sensible heat derived from eddy covariance, satellite, and meteorological observations. J. Geophys. Res.: Biogeo., 116, G00J07, http://dx.doi.org/10.1029/2010JG001566, 2011.

Orth, R., and Destouni, G: Drought reduces blue-water fluxes more strongly than green-water fluxes in Europe. Nat. Commun., 9, 3602, http://dx.doi.org/10.1038/s41467-018-06013-7, 2018.

Vicente-Serrano, S. M., Van Gerard, V. D. S., Beguería, S., Azorin-Molina, C., and Lopez-Moreno, J. I.: Contribution of precipitation and reference evapotranspiration to drought indices under different climates. J. Hydrol., 526, 42–54. http://dx.doi.org/10.1016/j.jhydrol.2014.11.025, 2015.

[Figure]

**Figure 1.** Data summary of the flux towers used in this study.

[Figure]

**Supplementary Figure S1.** Spatial distribution of (a) the flux towers in the FLUXNET2015 and (b) the weather stations used in this study. The plant function types of the flux towers include Croplands (CRO), Deciduous Needleleaf Forests (DNF), Evergreen Needleleaf Forest (ENF), Evergreen Broadleaf Forest (EBF), Deciduous Broadleaf Forest (DBF), Mixed Forest (MF), Grasslands (GRA), Savannas (SAV), Woody Savannas (WSA), Closed Shrublands (CSH), Open Shrublands (OSH), Wetlands (WET), and Snow and Ice (SNO).

[Figure]

**Supplementary Figure S2.** The performance of the ANN model using different number of neurons.

[Figure]

**Figure 2.** Density scatter plot for (a) the cross-validation in terms of values and (b) the cross-validation in terms of trends. The validation set of values cross-validation is randomly composed of 10 flux towers from different plant function types, and the validation set of trends cross-validation is composed of the trends calculated from all time periods.

[Figure]

**Supplementary Figure S5.** Density scatter plot for the cross-validation in terms of trends for different samples from ENF, EBF, DBF, MF, and OSH, respectively. The validation set is randomly composed of one flux tower from one plant function type, and the trends are estimated for all time periods.

[Figure]

**Supplementary Figure S6.** Density scatter plot for the cross-validation in terms of trends for different samples from SAV, GRA, CRO, and WET, respectively. The validation set is randomly composed of one flux tower randomly selected from one plant function type, and the trends are estimated for all time periods.

**Table S1. Test results of model training using different variable combinations***

| Combination of different variables | λE | | H | |
|---|---|---|---|---|
| | R | RMSE (W m$^{-2}$) | R | RMSE (W m$^{-2}$) |
| {Tmax; Tmin} | 0.60 | 36.22 | 0.52 | 43.75 |
| {RH; Tmax; Tmin} | 0.66 | 32.11 | 0.60 | 40.97 |
| {RH; Tmean; Tmax; Tmin} | 0.67 | 32.00 | 0.61 | 40.06 |
| {RH; Tmax; Tmin; DTR} | 0.67 | 30.89 | 0.62 | 39.48 |
| {SW_IN_POT; Tmax; Tmin} | 0.70 | 30.75 | 0.78 | 31.81 |
| {SW_IN_POT; RH} | 0.70 | 30.72 | 0.69 | 37.01 |
| {SW_IN_POT; RH; Tmean; Tmax; Tmin} | 0.74 | 28.80 | 0.72 | 35.8 |
| {SW_IN_POT; RH; Tmean; Tmax; Tmin; WS} | 0.75 | 28.65 | 0.74 | 34.72 |
| {SW_IN_POT; RH; Tmax; Tmin; WS} | 0.74 | 28.78 | 0.73 | 35.06 |
| {SW_IN_POT; RH; Tmax; Tmin; WS; P} | 0.75 | 28.90 | 0.75 | 33.01 |
| {SW_IN_POT; RH; Tmax; WS; Tmin; P} | 0.76 | 28.11 | 0.75 | 34.80 |
| {SW_IN_POT; RH; Tmean; Tmax; Tmin; DTR; WS; P} | 0.77 | 27.12 | 0.74 | 34.34 |

*Tmax, Tmin, and Tmean are maximum, minimum, and mean temperature, respectively. RH, DTR, and SW_IN_POT are relative humidity, daily temperature range, and top-of-atmosphere shortwave, respectively. WS and P are mean wind speed and total precipitation of the day.

**Table S2. Variables and data sources for training ANN model***

| Variables | Units | Data sources | Usage |
|---|---|---|---|
| SW_IN_POT | W/m$^2$ | The daily integrated dataset | Input variable |
| Tmean | °C | The daily integrated dataset | Input variable |
| Tmax | °C | Half-hourly or hourly data | Input variable |
| Tmin | °C | Half-hourly or hourly data | Input variable |
| VPD | hPa | The daily integrated dataset | VPD was used to calculate RH |
| WS | m/s | The daily integrated dataset | Input variable |
| λE | W/m$^2$ | The daily integrated dataset | Output variable |
| H | W/m$^2$ | The daily integrated dataset | Output variable |

*Vapor pressure deficit (VPD) was used to calculate relative humidity. λE and H are the latent heat flux and sensible heat flux, respectively.

---

## Author Comment (AC3) · 29 Jan 2021

Supplementary Table S1: To reduce time comsuption, here we used a simplified ANN structure with 2 hidden layers and 6 neurons per layer to perform the test of different variable combinations. Thus, its performance is different from the formal ANN model.

[Figure]

---

## Author Response (AR1)

**Response to the reviewers' comments**

**Response to Referee #1**

We greatly appreciate Dr. René Orth providing constructive comments and suggestions on our manuscript HESS-2020-590. Your comments help us improve the manuscript and also guide us to improve our study. We seriously considered all your comments and revised the manuscript accordingly. The individual comments are replied below. The comments are shown in black font and our responses are shown in blue font.

**Comment (1):** This study derives land evapotranspiration with a machine learning approach applied to daily meteorological station data from across the world. More decreasing than increasing trends are found across the Earth's land areas. The controls for these trends are determined and distinguished by jointly considering trends in evapotranspiration and precipitation minus evapotranspiration as a proxy for runoff.
* * *
Recommendation: I think the paper requires major revisions.

The methodology and research question addressed in this paper are novel and relevant, making it a potentially good fit for HESS. Also the joint consideration of evapotranspiration and (proxy) runoff to interpret the reported trends and associate them with potential causes is an important contribution to the land surface science community. However, before the paper is suitable for publication in HESS, some critical shortcomings need to be addressed.

**Reply:** Thanks a lot for your encouraging and constructive comments. We take each of them seriously in revising our manuscript. We have carefully revised the manuscript following your comments and suggestions, and hope that our revisions have satisfactorily addressed all your concerns and questions. Detailed responses to each specific comment can be found below.

**Comment (2):** I like that the authors validated their simulation results against observed streamflow and a reference ET product. However, the diagnosed agreement with these products is actually not very convincing, particularly in terms of the trends, as shown in Figure S8, and in the comparison between Figures 4c and S11. I think it is critical to understand these differences between the products, as otherwise I am missing convincing evidence that the data simulated here can be used to assess global ET and runoff trends.

**Reply:** We agree with the point that the differences between the products should be better understood. The difference between our ET and the ET estimated by the model tree ensemble

(MTE) can be caused by the different models, forcing data, and time scales. Meanwhile, we should not fully expect the P-ET to be completely consistent with observed streamflow, because the observed streamflow is strongly affected by human activities especially on long time scales. We also note that because our work is based on a boundary layer perspective which physics does not change in time. This is absed on Salvucci and Gentine (2015) and Gentine et al., (2016) who showed that the atmospheric boundary layer and tits diurnal cycle were indicators of the surface energy partitioning. Estimates of ET based on instantaneous weather data and land-surface properties (e.g. LAI) are affected by additional confounding factors that are not measured like nutrients or increased $CO_2$ whose trends cannot be captured and that will invalidate their trends. In addition, most products such as Fluxnet MTE or FLUXCOM rely on remote sensing data and therefore only have a relatively short record.

According to the comment, we added more discussions and expanded the cross-validation in terms of trends as suggested in the comment (3).

**Comment (3):** Adding to (1), it would be insightful to expand the cross-validation analysis from Figure 2 to validate the derived data also in terms of the trends observed at the independent cross-validation stations, as the final conclusions of this study are build on the trends rather than the short-term variability of the data.

**Reply:** Thanks for the suggestion. We expanded the cross-validation analysis in terms of trends in the revised manuscript. It was suggested that the trends predicted by the ANN model were highly correlated with the observed trends, and even in most cases, the estimation of the trends were more reliable than the estimation of the values (please see the Attached Figures at the end). The key revised parts are also copied here as follows:

*"As for the prediction of trends in latent heat and sensible heat fluxes, the ANN model also shows good performance (Fig. 2b). All correlation coefficients between the estimated λE (H) trends and observed λE (H) trends exceeded 0.90 (p<0.001) on ENF, DBF, GRA, and WET, and the correlations on MF, OSH, and CRO exceeded 0.80 (p<0.001), and the correlations surpassed 0.70 (p<0.001) on EBF and SAV (Supplementary Fig. S5 and Fig. S6). In most causes, the accuracy of λE (H) trend estimations using the ANN model are higher than the accuracy of individual λE (H) value estimations."*

**Comment (4):** The comparison of ET and runoff trends between the CMIP5 scenario simulations and the machine learning-derived, historical simulations, does not really make sense as totally different time periods are considered to compute the respective trends, if I get this right?

**Reply:** Yes, the trends vary from different time periods. We acknowledge that a consistent time period is better, but it is difficult to achieve this due to the paucity of data. Moreover, the main purpose here is to identify the change directions of ET or runoff rather than their magnitudes. Thus, we can determine whether the trends of decrease (increase) in ET or runoff is a long-term existing phenomenon by comparing the change directions over different time periods in history and in simulated future scenarios.

After considering the comment, we have added an explanation in section 3.3.

**Comment (5):** The description of the employed machine learning algorithm is not clear. The choice of artificial neural networks over other machine learning methods is not sufficiently motivated. Why is it more suitable than for example random forests in this context?

**Reply:** Different machine learning algorithms have their own advantages. The artificial neural networks tend to work well, as it has strong nonlinear function approximation capability and fault tolerance. As our selected neural networks demonstrated good performance, to save space, we did not compare different machine learning algorithms in this study. Since we are dealing with very shallow networks there should be only limited difference between ML models.

We still considered your comments and added a description on in section 2.4 as follows:
*"The artificial neural networks (ANN) have been shown to be powerful non-linear regressions tools. Pure ANN models have good performance in retrieving surface fluxes, and even in some cases, its performance is better than the hybrid model"*

**Comment (6):** Why not simply use FLUXCOM instead of deriving yet another estimate?

**Reply:** The data in the FLUXCOM or the results estimated by the model tree ensemble (MTE) are driven by remote-sensing and instantaneous meteorological observations, and thus they rely on the satellite era and cannot be used for long-term trends as they cannot represent the long-term effects of confounders such as $CO_2$ or nutrients or species composition change. This is why we use an opposite view – we use in essence a boundary layer budget based on Salvucci and Gentine (2015) and Gentine et al. (2016) except that we lump the non-linearity for the boundary layer budget in a neural network. The physics of the boundary layer is not changing in time and weather station have been available for many decades, allowing the development of the first very long-term record of hydrologic trends. Indeed, the diurnal cycle of temperature is directly related to the sensible heat flux. Similarly, the course of specific humidity related to the rate of latent heat flux. If there are changes in latent heat flux due for instance to stomatal closure to higher $CO_2$, this is still captured by the change in the specific humidity in the boundary layer. Finally,

FLUXNET-MTE and FLUXCOM are based on data that are highly localized, especially in the northern hemisphere. The network of weather station is both much longer but also extend to many locations such as in the tropics, providing much more constrain in those places, where other retrievals typically display very large uncertainties. The aim of our strategy in this study is therefore to infer a longer surface fluxes (as well as better generalization to the tropics and other remote regions). Thus, the length of the surface heat fluxes in existing products does not match with our research purpose. After considering this comment, we added an explanation in the introduction section.

The key points of the above response are also presented in the revised manuscript.

**Comment (7):** Further, the setup of the ANN model is unclear, i.e. how the hyper-parameters are chosen (why exactly 2 hidden layers? why 500 epochs?). Why different performance metrics (RMSE and MSE) are chosen, how the training is done, and how overfitting is prevented. I acknowledge that some of these choices are necessarily arbitrary, but this would be good to mention, including some tests on the relevance of these choices for the conclusions of the study.

**Reply:** The more hidden layers and neurons in the ANN, the stronger the nonlinear ability of the model, but the complexity and training time are also increasing. In theory, a neural network with 2 hidden layers can realize any complex mapping, as the nonlinear ability can be enhanced by adding neurons. The ANN model with 2 hidden layers and 15 neurons shows good performance and appropriate time consumption (see the attached Fig. S2) As for the optimal number of neurons, we initially determined it according to the empirical formula:

$$h = \sqrt{(n+m)} + a$$

Where $n$ is the number of input neurons, and $m$ is the number of output neurons, and $a$ is a constant ranging from 0 to 10. The detailed revised contents in the revised manuscript are also copied here as follows:

*"A neural network with 2 hidden layers can achieve the same performance as with a large number of hidden layers, so we used the lowest complexity model and enhanced its nonlinear ability by adding neurons. As for the optimal number of neurons, we initially tested it according to an empirical formula, i.e., $h = \sqrt{(n+m)} + a$ (n is the number of input neurons, m is the number of output neurons, and a is a constant ranging from 0 to 10). The neural network was determined to have two hidden layers and 15 neurons per hidden layer, and the ANN model showed good performance and appropriate training time (Supplementary Fig. S2)."*

MSE was an indicator used to evaluate the performance of neural works in the training process of adjusting weight, and RMSE is an indicator used to analyze the bias between the ANN predicted surface fluxes and the observed surface fluxes in validation set.

To avoid over-fitting, the early stopping method was used to avoid overfitting this study, that is, we recorded the best validation accuracy during the training process, and the training was stopped when the MSE was no longer reduced after going through the entire dataset. The detailed revised parts according to the comment are also copied here:

*"To avoid over-fitting, the early stopping method was used, that is, we recorded the best validation accuracy during the training process, and the training was stopped when the MSE was no longer reduced after going through additional epochs."*

We apologize for the omission of some descriptions on the setup of training the ANN, and we have added more information on the process in the section 2.4 accordingly.

**Comment (8):** There are many small language errors (such as missing articles or wrong grammar) throughout the manuscript. The authors need to take special care of these when revising the manuscript.

**Reply:** The language has been polished by a language editing agency, and a bilingual colleague, and we have taken special care for each sentence and add missing articles in the references.

**Comment (9):** lines 33 & 36: Please explain what you mean with "offline".

**Reply:** The calculation of potential evaporation in drought index using meteorological variables from climate model outputs is offline. We have modified this sentence to *"Using potential evaporation rather than actual ET or calculating offline ET using meteorological variables from climate model outputs in the traditional drought indices, the calculation implicitly assumes that soil can always supply moisture to meet the atmospheric evaporation demand"*.

**Comment (10):** lines 57, 72, 83, 123/124: You mention in these places different sets of variables which are (not?) used by the ANN algorithm, please clarify.

**Reply:** ①Top-of-atmosphere shortwave radiation, relative humidity, temperatures (mean, maximum, and minimum temperatures), and surface wind speed are the variables used by the ANN algorithm (in line 123/124).

②The expressions of temperatures, humidity, and solar radiation (in line 57) are a broad concept and does not refer to a specific variable. We have rephrased this sentence to "This

approach utilizes daily observations of meteorological variables such as temperatures, humidity, and solar radiation."

③The expressions of top-of-atmosphere shortwave radiation, vapor pressure deficit (VPD), mean temperature, and surface wind speed (in line 72) are referring to the data collected from the integrated daily product of FLUXNET2015. VPD is used to calculate relative humidity, and daily maximum and minimum temperatures are obtained from half-hourly/hourly flux tower measurements. These data are used to train the ANN model.

④The expressions of precipitation, temperatures (mean, maximum, and minimum temperatures), dew point temperature, and surface wind speed (in line 83) are referring to the data collected from weather stations. Dew point temperature was used to calculate relative humidity at weather stations, and the meteorological data collected from global weather stations were used to drive the trained neural network models.

According to the comment, we have modified the expressions and provided more information in the revised manuscript.

**Comment (11):** line 69: How is this gap filling done?

**Reply:** The gap-filling data were provided by the FLUXNET.

**Comment (12):** line 71: Why are you targeting daily resolution? To infer trends, monthly resolution might be sufficient?

**Reply:** We agree with that monthly scale might be sufficient for inferring trend, but the daily scale is required to assess the fluxes in the boundary layer as they are informed based on the diurnal cycle. There is no contradiction between retrieving surface fluxes on daily scale and aggregating on monthly scale.

**Comment (13):** lines 86-91: I do not fully understand this paragraph. Do the original and target station groups differ by the number of stations in some grid cells? If yes, do you average across multiple stations in one grid cell, or remove stations to only keep one (and which one?)?

**Reply:** Thanks for the comment, and we have improved the writing of this paragraph. The target stations were obtained according to the following three steps, i.e., (1) The stations with a time series spanning less than ten years were excluded; (2) if the stations had the same geographic coordinates, we used the stations with longer observation to replace the stations with shorter observation; (3) if there were multiple stations showing different coordinates in a 0.1-degree grid, we removed the stations with a shorter observation length.

**Comment (14):** line 101: Why do you employ top-of-atmosphere radiation instead of surface radiation which is what the vegetation is actually exposed to?

**Reply:** Because there do not exist reliable long-term surface observational solar radiation data, top-of-atmosphere shortwave radiation is a more direct measurement. After considering this comment, we have added an explanation in section 2.3 and this is also copied here as follows:

*"Solar shortwave radiation is a key factor affecting surface energy and water cycle. Since there is no reliable long-term surface observational solar radiation data, top-of-atmosphere shortwave radiation was used as a replacement. Cloud effects are inherently captured by the diurnal cycle of temperature and humidity (Gentine et al., 2013a,b).".*

**Comment (15):** lines 156/157: Where is the influence of the resistances shown, or how do you arrive to this conclusion?

**Reply:** We demonstrate the influence in an indirect way by inferring changes from a big leaf model (which is not used to derive the ANN). According to this comment and the Eq. (14), we have modified the sentences as following:

*"Therefore, EF is closely connected with surface resistance and aerodynamic resistance. A decline in EF can be induced by an increase in surface resistance. Annual EF ranges from 0 to 1, and $r_a$ is a function of wind speed with a relatively small variations while the variations in $r_s$ can be strong."*

**Comment (16):** line 160: How was the cross validation done? How/which stations or times have been chosen as independent validation data?

**Reply:** We randomized the samples of five flux towers (moderate number) from 212 sites as the validation set, and then used the remaining samples to train the ANN model. The ANN predicted daily $\lambda E$ (H) of the validation set were compared with their observed values. As for validation under different land covers, we randomly select samples from five flux towers under this land type as the validation set.

According to this comment, we have added a description of the random selection of cross-validation samples. Meanwhile, we have redrawn the Fig. 2 by randomly selecting 10 flux towers from different plant functional types as the validation set (see Fig. 2 in the attached Figures).

**Comment (17):** line 168: I would think that also SAV is water-limited?

**Reply:** Agree.

**Comment (18):** lines 165, 169: Why are there different correlations given for OSH?

**Reply:** Apology for the typo. We have corrected it in the revised manuscript.

**Comment (19):** line 184/185: What does 0.78~0.79 and 0.77~0.78 mean?

**Reply:** The range of correlation coefficients. We have corrected these sentences.

**Comment (20):** line 195: The reference for FLUXCOM would be Jung et al. 2019 which is also in the reference list, or did you actually use the MTE product from Jung et al. 2010?

**Reply:** Thanks for the reminder. We used the ensemble data of latent heat flux on land from the Department of Biogeochemical Integration (BGI) of the Max Planck Institute (MPI) (https://www.bgc-jena.mpg.de/geodb/projects/Data.php). The data were retrieved using the model tree ensemble (MTE) approach for upscaling FLUXNET measurements (Jung et al., 2011). We have corrected the reference and relevant expressions in the revised manuscript.

**Comment (21):** line 220: I cannot see a cooling trend in northern Europe in Figure 3.

**Reply:** We have corrected it in the revised manuscript.

**Comment (22):** lines 226-228: Phrasing could be improved, the "when" is not needed here.

**Reply:** We have rephrased this sentence.

**Comment (23):** line 235: Not sure if I would see a "persistent long-term trend" anywhere here as the spatial patterns of trends vary quite a lot across time periods in Figure 5.

**Reply:** We has revised the sentence to avoid inaccurate expression.

**Comment (24):** Which models did you consider in particular?

**Reply:** The simulation is an ensemble from Phase 5 of the Coupled Model Intercomparison Project (CMIP5) under the RCP8.5 scenario. We have modified the relevant expression.

**Comment (25):** line 261: There is no "RCP8.5 climate model".

**Reply:** Apology for the wrong expression, and we have corrected it accordingly.

**Comment (26):** section 3.4: Very nice approach and analysis to infer potential causes of the observed trends.

**Reply:** Thanks for the favorable evaluation.

**Comment (27):** Figure 1: What are the black empty bars which are superimposed on the colored bars?

**Reply:** Black empty bars represent the towers in the FLUXNET2015 Dataset and the solid bars represent all towers registered in FLUXNET.

We have redrawn the Figure 1 and its related Figure S1a to make the illustration clear (see Fig. 1 and Fig. S1 in the Attached Figures).

**Comment (28):** Figure 3: How is the Sahara desert defined? Why are other deserts not excluded, too? What is the time period over which the trends are computed? How is the spatial interpolation between the station locations done?

**Reply:** There is no strict definition of the Sahara desert. We prefer excluding the Sahara because it has the largest desert area and scarce meteorological observation data. We referred to some previous studies on drought without considering the Sahara region, such as Vicente-Serrano et al., 2015.

The trends are computed over the period of 1950-2017. If the observational data is not that long, the trends are converted to a uniform length of 68 years. The spatial interpolation in this study uses the Kriging interpolation method based on ArcGIS platform. We have revised the manuscript according the enlightening comment.

**Comment (29):** Figure S10: This is cool, but where is the information coming from?

**Reply:** Data are collected from the Food and Agriculture Organization of the United Nations (http://www.fao.org/nr/water/aquastat/irrigationmap/index10.stm). We have provided the data source in the revised manuscript.

**Comment (30):** Tables S1 and S2: Some of the variable names here need more explanation. And why was only the u-component of the wind speed used?

**Reply:** We have provided more explanations about the variable names in Tables S1 and S2 (see Attached Tables), and have improved the writing of this section.

The wind speed used in this study is mean surface wind speed of the day, not the u-component of the wind speed. We have modified the abbreviation of wind speed.

**References:**

Berg, A., Findell, K., Lintner, B., Giannini, A., Seneviratne, S. I., van den Hurk, B., Lorenz, R., Pitman, A., Hagemann, S., Meier, A., Cheruy, F., Ducharne, A., Malyshev, S., and Milly, P. C. D.: Land-atmosphere feedbacks amplify aridity increase over land under global warming. Nat. Clim. Change, 6, 869–874, https://doi.org/10.1038/nclimate3029, 2016.

Gentine, P., Chhang, A., Rigden, A., and Salvucci, G.: Evaporation estimates using weather station data and boundary layer theory. Geophys. Res. Lett., 43, 11661–11670, https://doi.org/10.1002/2016GL070819, 2016.

Gentine, P., Entekhabi, D., and Polcher, J.: The diurnal behavior of evaporative fraction in the soil-vegetation-atmospheric boundary layer continuum. J. Hydrometeorol., 12, 1530–1546, https://doi.org/10.1175/2011JHM1261.1, 2011.

Jung, M., Reichstein, M., Ciais, P., Seneviratne, S.I., Goulden, M. L., Bonan, G., Cescatti, A., Chen, J., de Jeu, R., Dolman, A. J., Eugster, W., Gerten, D., Gianelle, D., Gobron, N., Heinke, J., Kimball, J., Law B. E., Montagnani, L., Mu, Q., Mueller, B., Oleson, K., Papale, D., Richardson, A. D., Roupsard, O., Running, S., Tomelleri, E., Viovy, N., Weber, U., Williams, C., Wood, E., Zaehle, S., and Zhang, K.: Recent decline in the global land evapotranspiration trend due to limited moisture supply. Nat., 467, 951–954, https://doi.org/10.1038/nature09396, 2010.

Jung, M., Reichstein, M., Margolis, H. A., Cescatti, A., Richardson, A. D., Arain, M. A., Arneth, A., Bernhofer, C., Bonal, D., Chen, J., Gianelle, D., Gobron, N., Kiely, G., Kutsch, W., Lasslop, G., Law, B. E., Lindroth, A., Merbold, L., Montagnani, L., Moors, E. J., Pagpale, D., Sottocornola, M., Vaccari, F., and Williams, C.: Global patterns of land-atmosphere fluxes of carbon dioxide, latent heat, and sensible heat derived from eddy covariance, satellite, and meteorological observations. J. Geophys. Res.: Biogeo., 116, G00J07, http://dx.doi.org/10.1029/2010JG001566, 2011.

Miralles, D. G., Van, d. B. M. J., Gash, J. H., Parinussa, R. M., de Jeu, R. A. M., Beck, H. E., Holmes, T. R. H., Carlos Jiménez, C., Verhoest, N. E. C., Dorigo, W. A., Teuling, A. J., and Dolman, A. J.: El Niño – La Niña cycle and recent trends in continental evaporation. Nat. Clim. Change, 4, 122–126, https://doi.org/10.1038/nclimate2068, 2013.

Orth, R., and Destouni, G: Drought reduces blue-water fluxes more strongly than green-water fluxes in Europe. Nat. Commun., 9, 3602, http://dx.doi.org/10.1038/s41467-018-06013-7, 2018.

Salvucci, G. D. and Gentine, P.: Emergent relation between surface vapor conductance and relative humidity profiles yields evaporation rates from weather data. Proc. Natl. Acad. Sci. USA, 110, 6287–6291, https://doi.org/10.1073/pnas.1215844110, 2013.

Vicente-Serrano, S. M., Van Gerard, V. D. S., Beguería, S., Azorin-Molina, C., and Lopez-Moreno, J. I.: Contribution of precipitation and reference evapotranspiration to drought indices under different climates. J. Hydrol., 526, 42–54. http://dx.doi.org/10.1016/j.jhydrol.2014.11.025, 2015.

[Figure]

**Figure 1.** Data summary of the flux towers used in this study.

[Figure]

**Supplementary Figure S1.** Spatial distribution of (a) the flux towers in the FLUXNET2015 and (b) the weather stations used in this study. The plant function types of the flux towers include Croplands (CRO), Deciduous Needleleaf Forests (DNF), Evergreen Needleleaf Forest (ENF), Evergreen Broadleaf Forest (EBF), Deciduous Broadleaf Forest (DBF), Mixed Forest (MF), Grasslands (GRA), Savannas (SAV), Woody Savannas (WSA), Closed Shrublands (CSH), Open Shrublands (OSH), Wetlands (WET), and Snow and Ice (SNO).

[Figure]

**Supplementary Figure S2.** The performance of the ANN model using different number of neurons.

[Figure]

**Figure 2.** Density scatter plot for (a) the cross-validation in terms of values and (b) the cross-validation in terms of trends. The validation set of values cross-validation is composed of 10 flux towers randomly selected from different plant function types, and the validation set of trends cross-validation is composed of the trends calculated from all time periods.

[Figure]

**Supplementary Figure S5.** Density scatter plot for the cross-validation in terms of trends for different samples from ENF, EBF, DBF, MF, and OSH, respectively. The validation set is randomly composed of one flux tower from one plant function type, and the trends are estimated for all time periods.

[Figure]

**Supplementary Figure S6.** Density scatter plot for the cross-validation in terms of trends for different samples from SAV, GRA, CRO, and WET, respectively. The validation set is randomly composed of one flux tower randomly from one plant function type, and the trends are estimated for all time periods.

**Table S1. Test results of model training using different variable combinations\***

| Combination of different variables | λE | | H | |
|---|---|---|---|---|
| | **R** | **RMSE (W m⁻²)** | **R** | **RMSE (W m⁻²)** |
| {Tmax; Tmin} | 0.60 | 36.22 | 0.52 | 43.75 |
| {RH; Tmax; Tmin} | 0.66 | 32.11 | 0.60 | 40.97 |
| {RH; Tmean; Tmax; Tmin} | 0.67 | 32.00 | 0.61 | 40.06 |
| {RH; Tmax; Tmin; DTR} | 0.67 | 30.89 | 0.62 | 39.48 |
| {SW_IN_POT; Tmax; Tmin} | 0.70 | 30.75 | 0.78 | 31.81 |
| {SW_IN_POT; RH} | 0.70 | 30.72 | 0.69 | 37.01 |
| {SW_IN_POT; RH; Tmean; Tmax; Tmin} | 0.74 | 28.80 | 0.72 | 35.8 |
| {SW_IN_POT; RH; Tmean; Tmax; Tmin; WS} | 0.75 | 28.65 | 0.74 | 34.72 |
| {SW_IN_POT; RH; Tmax; Tmin; WS} | 0.74 | 28.78 | 0.73 | 35.06 |
| {SW_IN_POT; RH; Tmax; Tmin; WS; P} | 0.75 | 28.90 | 0.75 | 33.01 |
| {SW_IN_POT; RH; Tmax; WS; Tmin; P} | 0.76 | 28.11 | 0.75 | 34.80 |
| {SW_IN_POT; RH; Tmean; Tmax; Tmin; DTR; WS; P} | 0.77 | 27.12 | 0.74 | 34.34 |

*Tmax, Tmin, and Tmean are maximum, minimum, and mean temperature, respectively. RH, DTR, and SW_IN_POT are relative humidity, daily temperature range, and top-of-atmosphere shortwave, respectively. WS and P are mean wind speed and total precipitation of the day.

**Table S2. Variables and data sources for training ANN model\***

| Variables | Units | Data sources | Usage |
|---|---|---|---|
| SW_IN_POT | W/m² | The daily integrated dataset | Input variable |
| Tmean | °C | The daily integrated dataset | Input variable |
| Tmax | °C | Half-hourly or hourly data | Input variable |
| Tmin | °C | Half-hourly or hourly data | Input variable |
| VPD | hPa | The daily integrated dataset | VPD was used to calculate RH |
| WS | m/s | The daily integrated dataset | Input variable |
| λE | W/m² | The daily integrated dataset | Output variable |
| H | W/m² | The daily integrated dataset | Output variable |

*Vapor pressure deficit (VPD) was used to calculate relative humidity. λE and H are the latent heat flux and sensible heat flux, respectively.

**Response to Referee #2**

**Comment (1):** The present study investigated the ratio of latent heat flux to available surface energy (EF) using an ANN method and FLUXNET and meteorological station data, and reported that EF decreased on a fractional land surface, especially, it was accompanied by increased runoff (precip–et). The topic of the study falls into the scope of the HESS journal, and the conclusion is interesting. A minor revision is recommended before its publication.

**Reply:** We thank the reviewer for the careful review. We also appreciate your valuable comments and suggestions, which help us improving our study and the quality of the manuscript. We have revised the manuscript following your suggestions, and hope that our revision has satisfactorily addressed your concerns. Detailed responses to each specific comment can be found below.

**Comment (2):** Major concerns: a) The validation of the ANN method needs further clarifications. From lines 161-162, r ranged from 0.782 to 0.768, corresponding to a $R^2$ of 0.59-0.61. More clarifications are need to prove that such accuracy is acceptable. Probably, the authors could compare the accuracy obtained in the study with those in previous similar studies.

**Reply:** We thank the reviewer for this valuable comment. We have expanded the cross-validation in terms of not only values but also trends and found that the trends predicted by the ANN model were highly correlated with the observed trend, and even in most cases, the estimation of the trends were more reliable than the estimation of the values (see the Attached Figure at the end). Those statistics are the ones being reported. Meanwhile, we have provided more discussion here according to your suggestion. The key revised parts are also copied here as follows:

*"As for the prediction of trends in latent heat and sensible heat fluxes, the ANN model also shows good performance (Fig. 2b). All correlation coefficients between the estimated λE (H) trends and observed λE (H) trends exceeded 0.90 (p<0.001) on ENF, DBF, GRA, and WET, and the correlations on MF, OSH, and CRO exceeded 0.80 (p<0.001), and the correlations surpassed 0.70 (p<0.001) on EBF and SAV (Supplementary Fig. S5 and Fig. S6). In most causes, the*

*accuracy of λE (H) trend estimations using the ANN model are higher than the accuracy of individual λE (H) value estimations."*

**Comment (3):** b) The writing of introduction and conclusion sections need to be improved. The introduction section: this section mainly stated that the traditional method did not consider the dynamic change of leaf stomatal resistance/conductance, while info about the similar studied based on observed data is a little bit limited. The conclusion section: this section is too simple, only the sentence starting with 'however' is a conclusion. Please add more info to this section.

**Reply:** Following the comment, we have extended discussion and comparison with existing observation-driven studies to illustrate the importance of our study in the introduction section. According to the suggestion for improving conclusion, we have improved and refined the conclusions in the revised version. The revised version of this part in Introduction as the reviewer mentioned is also copied here as follows:

"*Existing studies with respect to global surface fluxes inferred from flux tower observations, remote sensing products and reanalysis data, and statistical approaches, e.g., the results estimated by model tree ensemble (MTE), rely on the satellite era and instantaneous meteorological observations (Jung et al., 2010; Jung et al., 2011; Miralles et al., 2013). Thus, the existing products cannot be used for long-term trends as they cannot represent the long-term effects of confounders such as $CO_2$ or species composition changes. This is why we use an opposite view – we use in essence a boundary layer budget based on Salvucci and Gentine (2015) and Gentine et al. (2016) except that we lump the non-linearity for the boundary layer budget in a neural network. Indeed, the diurnal cycle of temperature is directly related to the sensible heat flux. Similarly, the course of specific humidity related to the rate of latent heat flux (Gentine et al., 2011). If there are changes in latent heat flux due to stomatal closure to higher CO2, this is still captured by the change in the specific humidity in the boundary layer. Moreover, the existing products are based on data that are highly localized, especially in the northern hemisphere. The network of weather station is both much longer but also extend to many locations such as in the*

*tropics, providing much more constrain in those places, where other retrievals typically display very large uncertainties. The aim of our strategy in this study is therefore to infer a longer surface fluxes as well as better generalization to the tropics and other remote regions.*"

**Comment (4):** Minor concerns: Line 18: 'namely that. . .' should be changed to 'namely, . . .'

**Reply:** Thanks a lot for the careful review. After considering the suggestion, we have modified 'namely that...' to 'that is,' in the revised manuscript.

**Comment (5):** Lines 35-38: the description about the ET output from models is incorrect, cause most simulated ET output actually has already considered insufficient soil moisture's influences.

**Reply:** We agree with this comment, and we have modified the expression of this sentence. We want to emphasize that some traditional drought assessments use the output of climate models (e.g., predicted temperature) to estimate potential ET. This is an offline method that can ignore insufficient soil moisture's influences. The revised sentence is also copied here as follows:

"*Using potential evaporation rather than actual ET or calculating offline ET using meteorological variables from climate model outputs in traditional drought indices, the calculation implicitly assumes that soil can always supply moisture to meet the atmospheric evaporation demand, which is an incorrect assumption for most land surfaces.*"

**Comment (6):** Line 64: 'EF' seems to appear for the first time here, so full name is needed for EF.

**Reply:** Done.

**Comment (7):** Line 72: radiation is missing after "shortwave"?

**Reply:** We have added the term "shortwave" in the revised manuscript.

**Comment (8):** Line 82: Station meteorological data might be a better caption.

**Reply:** Thanks a lot for the valuable comment. We have modified the caption to "Observed weather station data", which is consistent with the expression in the full text.

**Comment (9):** Line 135: 'rs' appears for the first time here? If so, the full name is needed.

**Reply:** Thanks a lot for the careful review. We have added the full name and avoid such omissions in the revised manuscript.

**Comment (10):** Line 242: References of Fu et al. (2012WRR, 2015JGR-A) are recommended here to illustrate ENSO's influences.

**Reply:** We thank the reviewer for recommending the useful references. Following this suggestion,we find that Fu et al (2012) is very suitable to illustrate the influences of ENSO after studying the papers. Thus, we have added this study as one of our references in the revised manuscript.

**References:**

Fu, C., James, A. L., and Wachowiak, M. P. Analyzing the combined influence of solar activity and El Niño on streamflow across southern Canada. Water Resour. Res., 48, W05507. https://doi.org/10.1029/2011WR011507, 2012.

Gentine, P., Entekhabi, D., and Polcher, J.: The diurnal behavior of evaporative fraction in the soil-vegetation-atmospheric boundary layer continuum. J. Hydrometeorol., 12, 1530–1546, https://doi.org/10.1175/2011JHM1261.1, 2011.

Gentine, P., Chhang, A., Rigden, A., and Salvucci, G.: Evaporation estimates using weather station data and boundary layer theory. Geophys. Res. Lett., 43, 11661–11670, https://doi.org/10.1002/2016GL070819, 2016.

Jung, M., Reichstein, M., Ciais, P., Seneviratne, S.I., Goulden, M. L., Bonan, G., Cescatti, A., Chen, J., de Jeu, R., Dolman, A. J.,, Eugster, W., Gerten, D., Gianelle, D., Gobron, N., Heinke, J., Kimball, J., Law B. E., Montagnani, L., Mu, Q., Mueller, B., Oleson, K., Papale, D., Richardson, A. D., Roupsard, O., Running, S., Tomelleri, E., Viovy, N., Weber, U., Williams, C., Wood, E., Zaehle, S., and Zhang, K.: Recent decline in the global land evapotranspiration trend due to limited moisture supply. Nat., 467, 951–954, https://doi.org/10.1038/nature09396, 2010.

Jung, M., Reichstein, M., Margolis, H. A., Cescatti, A., Richardson, A. D., Arain, M. A., Arneth, A., Bernhofer, C., Bonal, D., Chen, J., Gianelle, D., Gobron, N., Kiely, G., Kutsch, W., Lasslop, G., Law, B. E., Lindroth, A., Merbold, L., Montagnani, L., Moors, E. J., Pagpale, D., Sottocornola, M., Vaccari, F., and Williams, C.: Global patterns of land-atmosphere

fluxes of carbon dioxide, latent heat, and sensible heat derived from eddy covariance, satellite, and meteorological observations. J. Geophys. Res.: Biogeo., 116, G00J07, http://dx.doi.org/10.1029/2010JG001566, 2011.

Miralles, D. G., Van, d. B. M. J., Gash, J. H., Parinussa, R. M., de Jeu, R. A. M., Beck, H. E., Holmes, T. R. H., Carlos Jiménez, C., Verhoest, N. E. C., Dorigo, W. A., Teuling, A. J., and Dolman, A. J.: El Niño – La Niña cycle and recent trends in continental evaporation. Nat. Clim. Change, 4, 122 – 126, https://doi.org/10.1038/nclimate2068, 2013.

Salvucci, G. D. and Gentine, P.: Emergent relation between surface vapor conductance and relative humidity profiles yields evaporation rates from weather data. Proc. Natl. Acad. Sci. USA, 110, 6287 – 6291, https://doi.org/10.1073/pnas.1215844110, 2013.

**Attached Figures**

[Figure]

**Figure 2.** Density scatter plot for (a) the cross-validation in terms of values and (b) the cross-validation in terms of trends. The validation set of values cross-validation is composed of 10 flux towers from different plant function types selected by a random function, and the validation set of trends cross-validation is composed of the trends calculated from all time periods.

---

## Author Response (AR2)

**Responses to the comments on hess-2020-590**

Dear Editor,

We greatly appreciate you for your efforts and time on our manuscript "Long-term relative decline in evapotranspiration with increasing runoff on fractional land surfaces" (#hess-2020-590), and we also thank Dr. René Orth and the other reviewer for providing valuable comments and suggestions on our manuscript. Following those comments and suggestions, we have revised the manuscript carefully. We are pleased to submit the revised version for your consideration with the main revisions listed below:

1. We revised each sentence according to the points raised by the reviewer, and we also carefully double-checked the language of the entire manuscript.
2. We improved our manuscript after seriously considering each comment, and replied to the questions/doubts with providing more experimental information.

We hope that the revised version of our manuscript will satisfy both you and the reviewers. Should you have any queries on our manuscript, please don't hesitate to contact us. Thank you again for time and efforts on our manuscript.

Yours sincerely,

Dr. Ren Wang and Dr. Pierre Gentine (on behalf of the authors' team)

Email: wangr67@mail2.sysu.edu.cn (R.W.); pg2328@columbia.edu (P.G.)

No.1, Wenyuan Road, Xianlin University District, Nanjing 210023, China (R.W.);

New York, NY 10027, USA (P. G.)

**Response to the reviewer**

The paper has overall improved as the authors have addressed some of the concerns raised by me and the other reviewer. However, at the same time important issues remain unresolved:

**Comment (1):** As pointed out in main comment (1) of my previous review, the validation of the inferred time series against independent data such as the Fluxnet-MTE product, and observed streamflow is not convincing. The differences are not satisfactorily explained by the authors in their rebuttal. Figure S11 clearly shows diverging trends in the tropics, and the author's explanation about "different models, forcing data and time scales" does not provide evidence why their product (and trends therein) is more reliable than the Fluxnet-MTE reference product. Moreover, the different runoff trends between Figures 4c and S14 cannot be entirely explained with human influence on the observed streamflow, I think, as even the large-scale patterns differ clearly, for example in Europe and Australia. Given that the differences between the products here are partly so remarkable I feel that they need to be understood better in order to illustrate the reliability of the newly derived dataset.

**Reply:** We thank the reviewer for this valuable comment. (1) We believe that the more important significance of our study is that it can provide an alternative choice of latent heat and sensible heat fluxes, which have its unique feature that it is purely driven by observations and can meet the needs for long-term trend analysis. Given that the Fluxnet-MTE assumes stationarity in the relationship between fluxes and environmental drivers (e.g. $CO_2$ effects are not taken into account), a direct comparison is difficult. In fact, this gap in the study of long-term trends with current products was the very reason behind our new product. Yet, we acknowledge that the well-trained ANN model and the retrieved results of our study can be an important supplement to the existing product.

(2) We provided more experimental information on the global distribution of large reservoirs which is related to the key impacts from human activities, e.g., reservoir scheduling and agriculture irrigation (Lehner et al., 2011) (Supplementary Fig. S16b). As suggested by the reviewer, Australia and Europe are regions where the observed runoff shows a decrease trend and is contrary to the retrieved P-ET trend. These regions are also distributed with a

number of large reservoirs. When we only considered the stations that are not too influenced by large reservoirs, we found that the direction and the spatial pattern of P-ET trend (Fig. 4c) are more obvious consistent with observed runoff trend, including the upward trend in northern Australia and the downward trend in southern Australia, the upward trend in western Europe and the downward trend in eastern Europe (Supplementary Fig. S16c). The spatial pattern of P-ET trends and observed runoff trends are also generally consistent in other regions including North and South America, Southern Africa, East Asia, and Southeast Asia.

[Figure]

**Supplementary Figure S16.** (a) Trends in observed runoff, (b) global distribution of large reservoirs, and (c) trends in runoff that are not too influenced by reservoir. Observed runoff data were collected from hydrologic stations with a controlled watershed area from 5 to 1000

km$^2$ with a data length over 20 years in the GRDC database. The right panel in (a) shows runoff trends at different latitudes, and the pink curve represents the median trend. The data of large reservoirs in (b) were collected from GRanD database, and the red dashed boxes show typical areas that the observed runoff presents a decrease trend which is opposite to the increase trend of P-ET. The observed runoff in (c) is selected by the following criteria: (1) 1 latitude/longitude degree away from the reservoirs with a capacity of <1 km$^3$ and (2) 5 latitude/longitude degree away from the reservoirs with a capacity of >1 km$^3$.

Lehner, B., Liermann, C. R., Revenga, C., Vörösmarty, C., Fekete, B., Crouzet, P., Döll, P., Endejan, M., Frenken, K., Magome, J., Nilsson, C., Robertson, J. C., Rodel, R., Sindorf, N., and Wisser, D.: High-resolution mapping of the world's reservoirs and dams for sustainable river-flow management. Front. Ecol. Environ., 9: 494–502. https://doi.org/10.2307/23034466, 2011.

**Comment (2):** As pointed out in main comment (2) of my previous review, different time periods are considered during which trends from the data derived in this study are determined versus the trends from the CMIP5 data. The authors acknowledge that matching time periods would be "better, but difficult to achieve", and state that this might not be so important as the evaluation focuses on directions of change rather than the actual magnitude. I do not agree with this. Even the sign of change may be different in different time periods, as for example introduced by decreasing and later increasing radiation within the global dimming and brightening periods. Further, decadal climate variability can introduce varying trends over time. Therefore, I think that with using different time periods, the comparison of trends between the data derived in this study and the CMPI5 data is not meaningful.

**Reply:** Thanks for the considerable comment. The prediction results of CMIP5 are auxiliary material for analyzing the phenomenon found in this study. After considering this comment, we placed the Fig. 6 to the supplementary material in the revised manuscript. We think the more important purpose here is to check the trend of EF changes, and the meaning of these comparisons lies in examining whether ET or EF show the similar trends in history and in the future $CO_2$ rising scenario.

**Comment (3):** Even though I requested this in main comment (4) of my previous review, the

authors did not explain why they chose to employ artificial neural networks over alternative machine learning approaches.

**Reply:** We thank the reviewer for this valuable comment. We added an comparison with another machine learning algorithm, i.e., random forest (RF). The RF model was trained for predicting daily λE and H based on the same Fluxnet2015 data as the ANN model. The RF model showed very similar performance to the ANN model. The correlation coefficients of RF model in predicting daily λE and daily H are 0.777 (p<0.001) and 0.756 (p<0.001), respectively (Supplementary Fig. S3). The correlation coefficients of ANN model in predicting daily λE and daily H are 0.849 (p<0.001) and 0.743 (p<0.001), respectively (Fig. 2). We want to point out that our retrievals are based on daily scale and the existing studies are based on monthly scale, and thus the values of correlation analysis can be different. Therefore, it is feasible to use the neural network algorithm to retrieve the latent heat and sensible heat fluxes.

[Figure]

**Supplementary Figure S3.** The performance of a well-trained random forest (RF) model for predicting daily λE and daily H based on the same Fluxnet2015 data as the ANN model. The RF model is trained using 500 trees. The test samples are randomly selected from the training data using a ratio of 10%.

The following explanations have been added in the revised manuscript and were copied here:

"The artificial neural networks (ANN) have been shown to be powerful regression algorithm, and unlike other machine learning algorithms, ANN can build multi-layer and multi-node network models to achieve deep learning of a complex simulation. Pure ANN

model has been proven to show good performance in retrieving surface fluxes (Chen et al., 2020; Haughton et al., 2018; Zhao et al., 2019)." (In the section 2.4)

"Moreover, we trained a random forest (RF) model for predicting daily $\lambda E$ and H based on the same Fluxnet2015 dataset as the ANN model. The RF model shows very similar performance to the ANN model. The correlation coefficients of RF model in predicting daily $\lambda E$ and daily H are 0.777 ($p<0.001$) and 0.756 ($p<0.001$), respectively (Supplementary Fig. S3). Therefore, it is feasible to use the neural network algorithm to retrieve the latent heat and sensible heat fluxes." (In the section 3.1)

**Comment (4):** Even though I mentioned this in main comment (5) of my previous review, there are still many small language errors throughout the manuscript, for example in lines 71, 77, 131, 155, 159, 196/197, 205, 224, and 249.

**Reply:** Thanks for the careful review. We have carefully double-checked the language in those lines and all expressions in the manuscript.

**Comment (5):** One additional issue caught my attention this time, sorry for realizing this only now. The ANN model is trained with target data from flux tower sites which typically cover 5-15 years. I doubt that CO2 in such short time periods would change sufficiently to lead to an emerging influence of CO2 fertilization on ET or EF. Therefore I think the reasoning of the CO2 fertilization effect on stomatal conductance potentially explaining differences in the partitioning between ET and runoff over time, which can be found throughout the manuscript, is not valid. I do not wish to remain anonymous - René Orth.

**Reply:** We appreciate this valuable point raised here. The effect of $CO_2$ fertilization dynamically affects the leaf area and physiology of vegetation, thereby changing the partitioning of latent heat and sensible heat fluxes. We completely agree with the reviewer for FLUXNET-MTE but not for our model. A major advantage of our method compared to other methods (e.g. FLUXNET-MTE) is that it does not rely on any assumption on a $CO_2$ effect on the link between environmental variables and fluxes. Indeed, we flipped the strategy around its head by diagnosing the diurnal changes in temperature and humidity in the boundary layer. As such this diurnal cycle reflects naturally any change in $CO_2$. For instance, if stoma were to

substantially close they would increase the sensible heat flux and reduce latent heat flux. This would in turn lead to increased temperature diurnal range and reduced atmospheric humidity in the boundary layer (Salvucci and Gentine 2013, Gentine et al. 2016). Therefore, this $CO_2$ effect is completely detectable with our method. We believe this is a major advantage of our method based on a boundary layer energy budget, as the physics of the boundary layer does not change (fluid dynamics).

Besides, according to Wang et al. (2020), which also used 22 flux towers with a time period over 14 years, the $CO_2$ fertilization effect has changed. This means that the observation period of the Fluxnet data used in this study is long enough.

Salvucci, G. D. and Gentine, P.: Emergent relation between surface vapor conductance and relative humidity profiles yields evaporation rates from weather data. Proc. Natl. Acad. Sci. USA, 110, 6287–6291, https://doi.org/10.1073/pnas.1215844110, 2013.

Gentine, P., Chhang, A., Rigden, A., and Salvucci, G.: Evaporation estimates using weather station data and boundary layer theory. Geophys. Res. Lett., 43, 11661–11670, https://doi.org/10.1002/2016GL070819, 2016.

Wang, S. H., Zhang, Y. G., Ju, W. M., et al: Recent global decline of $CO_2$ fertilization effects on vegetation photosynthesis. Science, https://doi.org/10.1126/science.abb7772, 2020.

**Comment (6):** lines 66-67: not clear what is meant with "boundary layer budget" and "non-linearity"

**Reply:** The "boundary layer budget" has been revised to "boundary layer energy budget" and the "non-linearity" has been revised to "non-linear effects of changing environment factors on surface energy fluxes". (Line 65-67).

**Comment (7):** lines 70-74: The authors state that existing datasets are somewhat biased towards the (flux) data availability which is largely concentrated in the northern hemisphere. This is true, but it is also true or the data derived in this study even if the weather stations are more distributed, because the model training is confined to the flux tower sites, as for the existing products.

**Reply:** Thanks for the comment. We agree with the point raised by the reviewer here, but the purpose here is to emphasize the weather station network are more widely distributed, ground-based observation, and long-term availability. One thing we want to emphasize again is that the physics of he boundary layer is consistent across regions unlike the flux to environmental response (which varies for instance with plant functional type, soil type etc). As such, the dataset we have is more representative of generic and global conditions. After considering this comment, we have modified the related expressions that existing products are based on highly localized data.

**Comment (8):** lines 156-157: Why was exactly this combination of input variables chosen while other combinations showed even better performance in Table S1, for example after including precipitation?

**Reply:** The combination of variables is based on the various environmental factors that reflect the changes in latent heat and sensible heat fluxes and the test results of the model training. When the training includes precipitation, the model shows a better performance in Table S1, but the improvement is limited and the signal of precipitation can be reflected in the variation of air humidity. In addition, latent heat flux measurements can be biased low during precipitation events (dew). As clarified in the Supplementary Materials (Page 13), to avoid overlapping information between the runoff assessed by P-ET and the retrieved ET, P is not included.

**Comment (9):** lines 165-166: What do you mean with initial tests and how did that influence the final decision? How was this formula applied to guide these decisions?

**Reply:** We meant that this empirical formula can provide a reference for us to determine a good neuron number when training our ANN model, and it can reduce the possibility of overfitting.

**Comment (10):** lines 204-213: Why are there different p-value thresholds used here?

**Reply:** The p-value is determined by the significance level of the correlation, and thus p-value of correlation analysis can be different. The statistical significance levels 0.001 and 0.05 are commonly used critical values.

**Comment (11):** line 220: Why >12000?

**Reply:** This is the magnitude of the sample number for correlation analysis. The value of the correlation coefficient can be affected by the number of samples.

**Comment (12):** lines 220-225: How are these trends computed? As linear trends? Over which time period?

**Reply:** The trends are computed using linear trend estimation. The time period is determined by the availability of the flux tower observed latent heat and sensible heat fluxes.

**Comment (13):** line 241: Why are you using the 11-year old Fluxnet-MTE product from Jung et al. 2010 rather than the more recent Fluxcom product from Jung et al. 2019?

**Reply:** The Fluxnet-MTE is a mature and widely applied machine learning product that can be used as a bench work. The MTE is based on only one machine learning method trained on monthly flux data and may be regarded as a precursor to FLUXCOM which comprises 147 products in two setups: (1) $0.0833^o$ resolution using MODIS remote sensing data (RS) and (2) $0.5^o$ resolution using remote sensing and meteorological data (RS + METEO).

As our study only use one machine learning method and one data-driven source (weather station observations), which is more similar to the data-driven methods of Fluxnet-MTE, so we use Fluxnet-MTE as a reference. We also want to point out that when we started this analysis FLUXCOM was not readily available (2018).

**Comment (14):** line 298: Which CMIP5 models are used in the considered ensemble, and how were their simulations aggregated?

**Reply:** We used data from 6 models: bcc-csm1-1, CanESM2, CESM1-BGC, GFDL-ESM2M, HadGEM2-ES, NorESM1-ME. Most of the models only have one ensemble member

available, so that we consider only one ensemble member per model. These detailed information have been provided in the revised manuscript.

**Comment (15):** Figures 3-7: How did you determine the area of the Sahara? References: Jung, M., et al. The FLUXCOM ensemble of global land-atmosphere energy fluxes, Scientific Data 6, 74 (2019). Jung, M., et al. Recent decline in the global land evapotranspiration trend due to limited moisture supply. Nature 467, 951–954 (2010).

**Reply:** As stated in the revised manuscript, the area of the Sahara is based on Vicente-Serrano et al., (2015). We have provided this information when it first appeared (Fig. 3).

[revised manuscript text omitted]

**Supplementary Figure S15.** Annual changes in (a) EF, (b) ET, and (c) P−ET in Phase 5 of the Coupled Model Intercomparison Project (CMIP5) under the RCP8.5 scenario with all anthropogenic forcing (e.g., land use/land cover changes, aerosols, and ozone). Changes are quantified by difference in years 2070−2099 of simulation and years 1941−1970. The data are collected from 6 models: bcc-csm1-1, CanESM2, CESM1-BGC, GFDL-ESM2M, HadGEM2-ES, NorESM1-ME. Most of the models only one ensemble member is available, so that we consider only one ensemble member per model.

[Figure]

**Supplementary Figure S16.** (a) Trends in observed runoff, (b) global distribution of large reservoirs, and (c) trends in runoff that are not too influenced by reservoir. Observed runoff data were collected from hydrologic stations with a controlled watershed area from 5 to 1000 km$^2$ with a data length over 20 years in the GRDC database. The right panel in (a) shows runoff trends at different latitudes, and the pink curve represents the median trend. The data of large reservoirs in (b) were collected from GRanD database, and the red dashed boxes show typical areas that the observed runoff presents a decrease trend which is opposite to the increase trend of P−ET. The observed runoff in (c) is selected by the following criteria: (1) 1 latitude/longitude degree away from the reservoirs with a capacity of <1 km$^3$ and (2) 5 latitude/longitude degree away from the reservoirs with a capacity of >1 km$^3$.

**Model training using different variable combinations**

Solar radiation is a key factor affecting surface energy and water cycle. However, there is currently a lack of long-term global ground shortwave radiation observation data. Thus, this study uses daily top-of-atmosphere shortwave (SW_IN_POT) which can be estimated by a semi-empirical model as the input of the artificial neural networks (ANN) model. This study used different variable combinations as the input of ANN model to analyze the sensitivity of different variables to the estimation of latent heat flux (λE) and sensible heat flux (H). To reduce time consumption, here we used a simplified ANN structure with 2 hidden layers and 6 neurons per layer to perform the test of different variable combinations. The results show that SW_IN_POT, relative humidity (RH), maximum temperature (Tmax), and minimum temperature (Tmin) are the key influencing variables, while mean temperature (Tmean) and mean wind speed (WS) are also important influencing variables (Table S1). Overall, λE is sensitive to SW_IN_POT, RH, and WS, and H is sensitive to SW_IN_POT, Tmax, and Tmin. When the model training includes precipitation, the model shows a better performance in Table S1, but the improvement is limited and the signal of precipitation (P) can be reflected in the variations of relative humidity. In addition, latent heat flux measurements can be biased low during precipitation events (dew). To avoid overlapping information between the runoff assessed by P minus evapotranspiration (ET) and the ET, P is not used as a driving variable of the ANN model. Meanwhile, daily temperature range (DTR) has overlapping information with Tmax and Tmin, and thus it is not used to train the ANN model. All variables and their data sources for training ANN model are listed in Table S2.

**Table S1. Test results of model training using different variable combinations***

| Combination of different variables | λE | | H | |
|---|---|---|---|---|
| | R | RMSE (W m$^{-2}$) | R | RMSE (W m$^{-2}$) |
| {Tmax; Tmin} | 0.60 | 36.22 | 0.52 | 43.75 |
| {RH; Tmax; Tmin} | 0.66 | 32.11 | 0.60 | 40.97 |
| {RH; Tmean; Tmax; Tmin} | 0.67 | 32.00 | 0.61 | 40.06 |
| {RH; Tmax; Tmin; DTR} | 0.67 | 30.89 | 0.62 | 39.48 |
| {SW_IN_POT; Tmax; Tmin} | 0.70 | 30.75 | 0.78 | 31.81 |
| {SW_IN_POT; RH} | 0.70 | 30.72 | 0.69 | 37.01 |
| {SW_IN_POT; RH; Tmean; Tmax; Tmin} | 0.74 | 28.80 | 0.72 | 35.8 |
| {SW_IN_POT; RH; Tmean; Tmax; Tmin; WS} | 0.75 | 28.65 | 0.74 | 34.72 |
| {SW_IN_POT; RH; Tmax; Tmin; WS} | 0.74 | 28.78 | 0.73 | 35.06 |
| {SW_IN_POT; RH; Tmax; Tmin; WS; P} | 0.75 | 28.90 | 0.75 | 33.01 |
| {SW_IN_POT; RH; Tmax; WS; Tmin; P} | 0.76 | 28.11 | 0.75 | 34.80 |
| {SW_IN_POT; RH; Tmean; Tmax; Tmin; DTR; WS; P} | 0.77 | 27.12 | 0.74 | 34.34 |

**Table S2. Variables and data sources for training ANN model***

| Variables | Units | Data sources | Usage |
|---|---|---|---|
| SW_IN_POT | W/m$^2$ | The daily integrated dataset | Input variable |
| Tmean | ºC | The daily integrated dataset | Input variable |
| Tmax | ºC | Half-hourly or hourly data | Input variable |
| Tmin | ºC | Half-hourly or hourly data | Input variable |
| VPD | hPa | The daily integrated dataset | VPD was used to calculate RH |
| WS | m/s | The daily integrated dataset | Input variable |
| λE | W/m$^2$ | The daily integrated dataset | Output variable |
| H | W/m$^2$ | The daily integrated dataset | Output variable |

*Vapor pressure deficit (VPD) was used to calculate relative humidity. λE and H are the latent heat flux and sensible heat flux, respectively.

---

## Author Response (AR3)

**Responses to the comments on hess-2020-590**

Dear Editor,

We thank you for your efforts and time on our manuscript (#hess-2020-590), and we also acknowledge Dr. René Orth for his careful review and enlightening comments. We are pleased to submit a revised version for your consideration. In addition to making changes based on the reviewer's comments, we also double-checked the manuscript.

We hope that the revised version will satisfy you. Should you have any queries on our manuscript, please don't hesitate to contact us. Thank you for your time and efforts again.

**Reply to Dr. René Orth**

Third review of Wang et al. "Long-term relative decline in evapotranspiration with increasing runoff on fractional land surfaces". I appreciate the efforts of the authors who have satisfactorily addressed most of the comments from my previous review. Below I list a few remaining minor points.

- regarding comment (1) (as numbered in the rebuttal file):

In the main text it should be Lehner et al. 2011 instead of 2017, in addition the doi in the reference is wrong.

**Reply:** Thanks for the careful review. We have corrected the year and the doi in the revised manuscript.

- regarding comment (2):

This sounds good but needs to be reflected and toned down correspondingly in the manuscript text.

**Reply:** We have modified the corresponding expression in the manuscript (section 3.3).

- regarding comment (3):

This is nice. Why not reproduce Figure 6 with the results of the implemented RF model
to study to illustrate the robustness of the main findings?

**Reply:** Thanks for this enlightening comment. Comparison of different models is not the focus of this study, and related content will be presented in our another manuscript. We do not redundantly illustrated the content here to avoid repetition.

- regarding comment (5):

The final sentence in the author's reply on the Wang et al. study addresses my comment,

this reasoning should be included in the main text.

**Reply:** Following this comment, we have added this sentence in the main text (Section 2.1).

- regarding comment (11):

You apparently removed the statement ">12000" but this is not documented in the track changes file. For me as a reviewer it is important that the track changes file really shows all additions and removals to the text. I do not wish to remain anonymous - René Orth.

**Reply:** We are sorry for our carelessness here. All changes can be tracked in this version of the manuscript.

[revised manuscript text omitted]